# Neural circuit selective for fast but not slow dopamine increases in drug reward

Peter Manza [1] ✉, Dardo Tomasi [1], Ehsan Shokri-Kojori[1], Rui Zhang [1], Danielle Kroll [1], Dana Feldman[1], Katherine McPherson [1], Catherine Biesecker[1], Evan Dennis [1], Allison Johnson[1], Kai Yuan[2], Wen-Tung Wang[3], Michele-Vera Yonga[1], Gene-Jack Wang [1] & Nora D. Volkow[1] ✉

The faster a drug enters the brain, the greater its addictive potential, yet the brain circuits underlying the rate dependency to drug reward remain unresolved. With simultaneous PET-fMRI we linked dynamics of dopamine signaling, brain activity/connectivity, and self-reported 'high' in 20 adults receiving methylphenidate orally (results in slow delivery) and intravenously (results in fast delivery) (trial NCT03326245). We estimated speed of striatal dopamine increases to oral and IV methylphenidate and then tested where brain activity was associated with slow and fast dopamine dynamics (primary endpoint). We then tested whether these brain circuits were temporally associated with individual 'high' ratings to methylphenidate (secondary endpoint). A corticostriatal circuit comprising the dorsal anterior cingulate cortex and insula and their connections with dorsal caudate was activated by fast (but not slow) dopamine increases and paralleled 'high' ratings. These data provide evidence in humans for a link between dACC/insula activation and fast but not slow dopamine increases and document a critical role of the salience network in drug reward.

Dopamine increases in brain reward regions are believed to underlie drug reward and to trigger the neuroplastic changes that result in addiction (i.e., moderate-to-severe substance use disorder)[1]. The rate at which dopamine increases along with the magnitude of the increase determine the intensity of a drug's rewarding effects and its addictive potential[2–5]. As such, routes of drug administration that result in the fastest delivery into the brain (e.g., intravenous injection and smoking) are more rewarding than routes that result in slower delivery (e.g., orally; with doses adjusted to reach equivalent levels in plasma)[2,6]. Compared to slower brain delivery, faster delivery of cocaine was associated with faster striatal dopamine increases, greater metabolic activity in brain reward circuitry, and more cocaine self-administration in rodents[4,7–10]. Individuals who smoke or inject drugs tend to develop substance use disorders more frequently and more severely, have higher overdose rates, and have more general health

problems than individuals who take drugs orally or by insufflation[11–14]. Further, the fastest routes of administration evoked the strongest self-reported pleasurable effects to IV cocaine (e.g., 'high', 'liking')[15,16]. These observations highlight the critical role of pharmacokinetics in drug reward and addiction potential. Therefore, understanding how the speed of drug delivery impacts human brain function and its association with reward could yield promising new targets for addiction treatment.

Preclinical studies have long used stimulants including methylphenidate (MP) as model drugs to study the relationship between pharmacokinetics and drug reward. But MP also has clinical relevance, for it is widely used to treat attention deficit hyperactivity disorder (ADHD). Because of its addictive potential[17] and clear evidence of misuse particularly when injected[18], MP is classified by the FDA as a schedule II substance alongside other addictive drugs like cocaine and

[1]National Institute on Alcohol Abuse and Alcoholism, National Institutes of Health, Bethesda, MD, USA. [2]School of Life Science and Technology, Xidian University, Xi'an, Shaanxi 710071, PR China. [3]Radiology and Imaging Sciences, Clinical Center, National Institutes of Health, Bethesda, MD, USA. ✉e-mail: peter.manza@nih.gov; nora.volkow@nih.gov

methamphetamine. Despite MP's addictiveness, it can be used safely and it is therapeutic when given orally. This is why MP formulations have been developed that make it harder to inject or snort MP (i.e., tamper-resistant formulations)[19].

Yet in humans there is surprisingly little data on how brain function changes based on stimulant drug pharmacokinetics. Early positron emission tomography (PET) studies showed that fast rises in striatal dopamine appear responsible for the feeling of 'high' to stimulants such as MP. Specifically, whereas oral MP produced the same total magnitude of striatal dopamine increases as IV administration, only IV MP induced a reliable experience of 'high' across healthy controls[20–22]. Similarly, despite equivalent levels of dopamine transporter blockade by cocaine in the brain of cocaine users, the rewarding effects were dependent on speed of delivery, with greatest 'high' reported for smoked cocaine (time to peak effects: 1.4 min), followed by intravenous (3.1 min) and then intranasal (14.6 min) routes of administration[23]. Still, the conscious experience of drug reward depends on more than local striatal responses, as dopamine signaling activates large-scale downstream networks via reciprocal cortical connections[24]. One such circuit that may be sensitive to dopamine dynamics comprises the nucleus accumbens and the ventromedial prefrontal cortex, which are strongly associated with drug reward[25,26], although other circuits may be equally implicated. For example, lesions to regions within the 'salience network' (most notably dorsal anterior cingulate cortex (dACC) and insula) can cause complete remission of addiction,[27,28] leading to hypotheses that this network underpins drug craving[29–31]. Thus, several candidate circuits may be sensitive to the speed of drug delivery and its association with drug reward.

To identify such a circuit, we used simultaneous PET-fMRI[32] while healthy adults received oral (resulting in slow brain delivery) and intravenous (resulting in fast brain delivery) doses of MP, in a double-blind, counterbalanced, randomized trial (Fig. 1a, b). We hypothesized distinct patterns of brain activity for oral and intravenous

administration, since slow dopamine increases primarily stimulate inhibitory $D_2$ receptors, which would decrease activity, whereas fast dopamine increases would additionally stimulate the low-affinity excitatory $D_1$ receptors, resulting in both increases and decreases in regional brain activation[33] (Fig. 1c). PET-fMRI and computational modeling studies in non-human primates and optical imaging studies in rodents have shown that $D_1$- and $D_2$-stimulation led to increases and decreases in the fMRI signal and in intracellular Ca signals, respectively[33–35].

## Results

For all subjects ($n = 20$), we first tested how cardiovascular (heart rate and systolic blood pressure) responses to oral MP (slow brain delivery), IV MP (fast brain delivery), and placebo differed in intensity and over time, using repeated measures drug condition × time ANOVA. The overall magnitude of systolic blood pressure, but not heart rate, was significantly affected by drug condition (main effect of drug: heart rate $F_{(2,2318)} = 2.746$, $p = 0.077$; systolic blood pressure $F_{(2,2277)} = 5.122$, $p = 0.011$). However, for both measures there was a significant drug condition × time interaction, as expected (heart rate $F_{(80,2318)} = 3.022$, $p < 2 \times 10^{-16}$; systolic blood pressure $F_{(80,2277)} = 3.403$, $p < 2 \times 10^{-16}$). Visual inspection showed that increases were strongest and fastest in the IV MP condition, whereas they were more modest and gradual in the oral MP condition, as expected (Supplementary Fig. 1).

Conventional 'static' analysis of PET imaging revealed significant decreases in relative standardized uptake value (SUVr) to oral (slow) and IV (fast) MP; these changes in [¹¹C]raclopride's specific binding to the drug administrations are a widely accepted measure of increases in synaptic dopamine concentrations (Fig. 2a). Henceforth we refer to these drug-induced decreases in [¹¹C]raclopride's specific binding as 'dopamine increases'. Non-displaceable binding potential (BPnd) was lower both for IV- and oral-MP relative to placebo for the 90 min scans (corrected $p < 0.05$). However, there was no significant BPnd difference between oral and IV MP ($F_{(1,85)} = 0.6$; $p = 0.44$; within-subjects

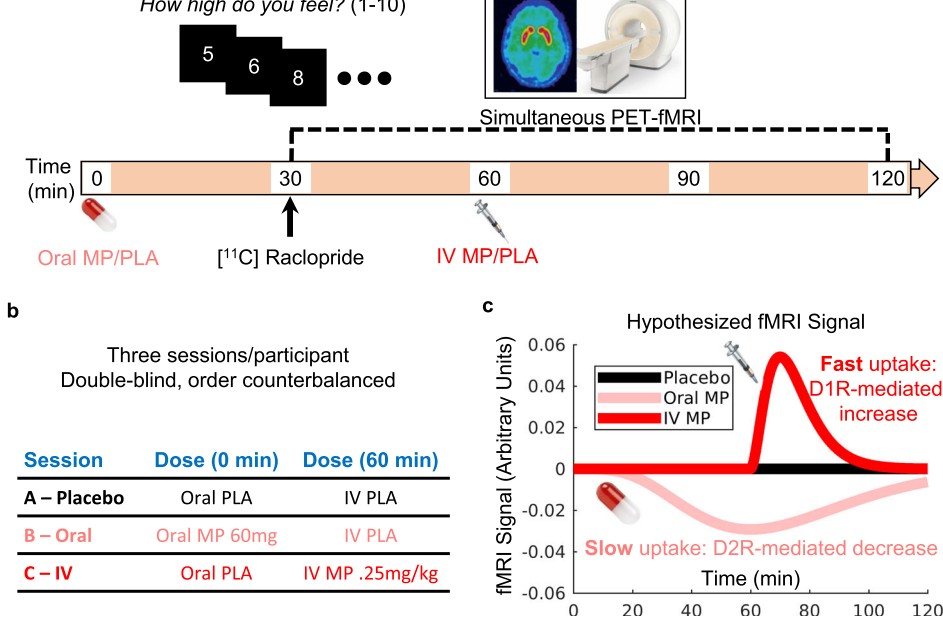

**Fig. 1 | Experimental design. a** Timeline of events. In each session, participants were given an oral dose of methylphenidate (MP) or placebo (PLA) at time 0; the [¹¹C] Raclopride bolus injection and simultaneous PET-fMRI scanning started at 30 min; an IV dose of MP or placebo was given at 60 min; and throughout the duration of the session participants used a button box in the scanner to self-report their experience of 'high' to the drug. **b** Session structure. Participants underwent three separate imaging sessions that were identical except for drug condition: Session A)

oral PLA and IV PLA (black color); Session B) oral MP (60 mg) and IV PLA (3 cc saline) (pink color); Session C) oral PLA and IV MP (0.25 mg/kg in 3 cc sterile water) (red color). **c** Hypothesized cortical-striatal fMRI signal from a simplified model based on postsynaptic dopamine receptor stimulation. We hypothesized opposing fMRI signal patterns to the IV (fast brain delivery) versus oral (slow brain delivery) MP doses, based on previous work[38].

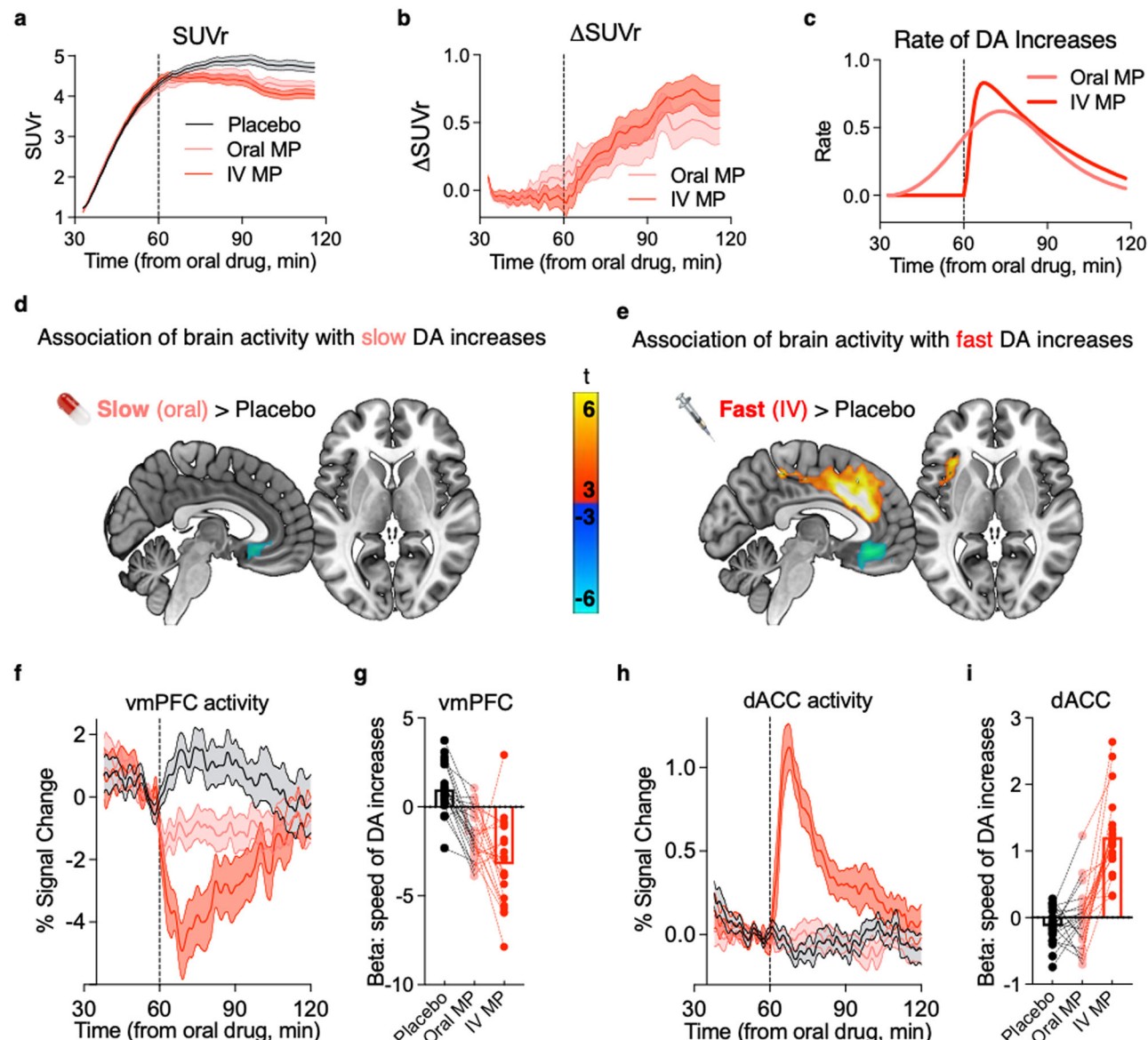

**Fig. 2 | Dynamic dopamine increases and associated brain activity to oral and intravenous (IV) methylphenidate (MP). a** Relative standardized uptake value (SUVr) showing the binding of [¹¹C]raclopride in the striatum for each drug condition. **b** Delta SUVr (i.e., Oral MP − Placebo and IV MP − Placebo) showing minute-by-minute differences in dopamine receptor occupancy to each drug administration. **c** The rate of dopamine increases to MP (derived from the derivative of the plot in panel b), which was used in subsequent analyses to estimate where brain activity paralleled speed of dopamine increases to MP. **d** Whole-brain analysis showing where brain activity was significantly associated with slow dopamine increases (oral MP) across time. **e** Whole-brain analysis showing where brain activity was significantly associated with fast dopamine increases (IV MP) across time. **f** Time courses of the BOLD fMRI signal extracted from the ventromedial prefrontal cortex (vmPFC) cluster that had significant decreases in activity to both oral and IV MP. **g** Beta values for each participant demonstrating the fit between

the time-course of vmPFC activity and speed of dopamine increases ($n = 20$ biologically independent adults). **h** Time-courses of the BOLD fMRI signal extracted from the dorsal anterior cingulate cortex (dACC) cluster that had increases in activity selective to IV MP. **i** Beta values for each participant demonstrating the fit between the time-course of dACC activity and speed of dopamine increases ($n = 20$ biologically independent adults). Data in panels (**a**) and (**b**) were previously reported[38]. In panels (**a**), (**b**), (**f**), and (**h**), the lines represent the mean of the 20 participants, and the shaded regions represent the standard error of the mean; the vertical dashed line denotes the time of the IV MP or placebo injection. The black color denotes the placebo session; pink denotes the oral MP session; and red denotes the IV session. fMRI time courses were temporally smoothed (for visualization only), and the y-axis units represent the percent signal change from the mean signal during the 'baseline' period, i.e., the first ten minutes at the beginning of the scan. Source data are provided as a Source data file.

ANOVA). Thus, there were no significant differences in the overall magnitude of dopamine increases between oral and IV sessions using this 'static' approach, which was expected as we used doses for oral and IV MP that were shown to occupy roughly equivalent quantities of striatal dopamine transporters (~70%) in humans[36,37].

However, we recently demonstrated that it is possible to resolve the dynamics of dopamine increases to MP with PET and [¹¹C]raclopride by taking the minute-by-minute difference in SUVr

between the placebo and MP conditions[38]. Dopamine increases to oral MP compared to IV MP started earlier (since oral MP was administered 30 min prior to [¹¹C]raclopride, whereas IV MP was administered 30 min post [¹¹C]raclopride) and were slower and more modest than the fast and strong increases from IV MP (Fig. 2b). The derivative of the fitted gamma cumulative distribution function to the average delta SUVr($t$) across subjects reflects the rate of striatal dopamine increases (Fig. 2c), which we used for subsequent

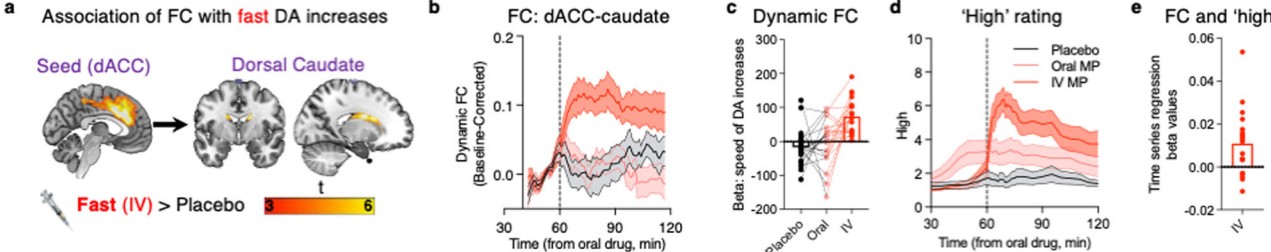

**Fig. 3 | Dynamic functional connectivity (FC) in a dorsal corticostriatal circuit is selectively associated with speed of fast, but not slow, dopamine increases and subjective 'high' to methylphenidate (MP). a** Dynamic brain FC in association with the speed of dopamine increases (as estimated by PET imaging), to intravenous (IV) MP. The significant dorsal anterior cingulate cortex (dACC) cluster from the analysis in Fig. 2e was used as a seed region. In voxelwise whole-brain analysis, the dorsal caudate emerged as the only region significantly positively connected with the dACC in association with speed of dopamine increases (negative clusters are also shown in Supplementary Fig. 8). Results from dACC are shown here, but the left insula also showed very similar significant patterns of dynamic FC with dorsal caudate (see Supplementary Fig. 9). **b** Time course of dynamic FC between dACC and dorsal caudate, for all three sessions. The lines represent the mean of the 20 participants, and the shaded regions represent the standard error of the mean. **c** Beta values for each participant demonstrating the fit between the time-course of dACC-to-dorsal caudate dynamic FC and speed of dopamine increases ($n = 20$ biologically independent adults). **d** Self-reported 'high' ratings for each drug condition, which were previously reported in another study using a portion of these data[38]. The lines represent the mean of the 20 participants, and the shaded regions represent the standard error of the mean. **e** Beta values for each participant demonstrating the fit between the time-course of dACC-to-dorsal caudate dynamic FC and subjective 'high' ratings ($n = 19$ biologically independent adults). The black color denotes the placebo session; pink denotes the oral MP session; and red denotes the IV session. Note: because in the oral MP session only 13 out of 20 participants reported feeling any 'high', we only show the results for the IV session for this analysis. Source data are provided as a Source data file.

analyses to identify brain circuits where activity coincided with dopamine dynamics.

Specifically, we examined where brain activity was significantly associated with dynamic dopamine increases across time. For the contrast of slow dopamine increases (oral MP > placebo), activity significantly decreased in ventromedial prefrontal cortex (Fig. 2d). For the contrast of fast dopamine increases (IV MP > placebo), activity also decreased in ventromedial prefrontal cortex, but it additionally increased in a large cluster comprising dorsal and middle anterior cingulate cortex and left insula (Fig. 2e; for one-sample t-tests see Supplementary Fig. 2; for cluster coordinates see Supplementary Table 2). Visual inspection of signals (BOLD time-courses) in these regions showed that, for ventromedial prefrontal cortex, activity decreased in a graded fashion (IV > oral > placebo) (Fig. 2f, g), whereas dACC and left insula had significantly higher activity only to fast dopamine increases (Fig. 2h, i; for insula cluster see Supplementary Fig. 3). A control analysis including amplitude of dopamine increases as a regressor in the model showed nearly identical results for rate of dopamine increases, and there were no significant clusters showing activity in association with dopamine amplitude (neither oral > placebo nor IV > placebo; Supplementary Figs. 4–5).

Next, we tested whether any of these significant clusters increased their connectivity with other brain regions in association with the speed of dopamine increases (i.e., we searched for circuits that synchronized with dopamine dynamics). We performed dynamic functional connectivity, using the aforementioned clusters as seed regions.

In paired t-tests, the ventromedial prefrontal cortex cluster did not have any significant dynamic connectivity patterns in association with slow nor fast dopamine increases. However, the left insula and the dACC showed significant functional connectivity with bilateral dorsal caudate in association with fast (IV MP) dopamine increases only (Fig. 3a–c; for one-sample t-tests, results from the insula seed functional connectivity analysis, and cluster coordinates, see Supplementary Figs. 6–9 and Supplementary Tables 3–4).

Finally, we sought to test whether the circuit identified from the prior analysis was significantly associated with individual differences in subjective drug reward. We first confirmed that 'high' ratings significantly differed in intensity and over time based on drug condition, using repeated measures drug condition × time ANOVA: (drug condition × time interaction: $F_{(80,2318)} = 10.30$, $p < 2 \times 10^{-16}$; Fig. 3d). Time-series regression analysis revealed a significant temporal association between each individual's dACC-dorsal caudate functional connectivity time-course with each individual's 'high' ratings time-course (IV session; $n = 19$ reporting feeling 'high'; $t_{(18)} = 3.115$; $p = 0.006$, Fig. 3e).

Although we were underpowered to formally test sex effects in the current study (11 males versus 9 females), we performed exploratory analyses for sex differences. Results from two-sample t-tests showed no significant sex effects for behavior (max high ratings); nor time-to-peak dopamine increases estimated from PET data; nor baseline binding potential (BPnd) in the nucleus accumbens, caudate, or putamen; nor strength of association between rate of dopamine increases and fMRI activations from all significant clusters identified in the manuscript, even at an uncorrected $p < 0.05$ threshold (all $p$'s > 0.22).

## Discussion

Together, these findings provide insight into how the salience network is critically linked to the pathophysiology of substance use disorder[28]. The dACC and bilateral insula responded to MP when administered intravenously, a route that maximizes the rewarding effects of drugs, but not when administered orally, the route used therapeutically that has much lower addiction potential. Consistently, these regions were among the most responsive, with a similar temporal course in the brain of cocaine users when given IV cocaine (0.6 mg/kg)[39]. The salience network, especially dACC, appears to be preferentially activated when the drug's route of administration leads to fast brain delivery and intense rewarding effects. The ACC performs numerous roles in cognition, emotion, and reward[40,41], which could all be relevant for processing the acute effects of stimulant drugs[39,42,43]. However, subregions of ACC are differentially involved in reward-related processes. In contrast to rostral ACC, which has been implicated in (inhibitory) affective processing, dACC, which had the most overlap with our findings, has been implicated in higher-order (drug independent) reward-based decision making but also in the inhibition of drug craving[44,45]. However, given that our experiment was performed in resting state without any task, we posit that dACC activation may reflect interoceptive signals mediating the experience of drug reward[46]. In fact, poor interoceptive awareness is a feature of substance use disorders and theorized to play a role in compulsive drug-seeking[47]. Moreover, the subregion of the ACC that was significant in our analyses corresponds to the one that underlies the connectivity

disruption associated with remission of addiction in patients suffering from brain lesions[28].

Crucially, after IV MP, the salience network had enhanced functional connectivity with the dorsal caudate, with which it has bidirectional connections[24,48]. The caudate receives extensive dopamine projections from the midbrain, and we previously reported that the speed of MP binding to the caudate strongly correlated with the experience of drug reward[12]. The current findings provide evidence that downstream cortical targets of the caudate appear necessary for the conscious experience of drug reward. This pattern may be driven by monosynaptic inputs from cortex to caudate, and/or by polysynaptic connections from caudate to cortex via pallidum and thalamus; future preclinical studies could resolve the circuit mechanisms in greater detail.

The ventromedial prefrontal cortex, a key node within the mesolimbic reward circuit, tracked both slow and fast dopamine pharmacokinetics, but its activity did not significantly relate to the experience of drug reward. The lack of an association could be because participants were naïve to stimulant drugs, a theory supported by a series of FDG-PET studies of brain glucose metabolism. In healthy adults, after a single dose of IV MP, metabolism in frontal cortex decreased, whereas after repeated doses it increased[49]. Further, the ventromedial prefrontal cortex was the only brain region differentiating healthy adults from cocaine users, as MP decreased metabolism in controls but increased it in cocaine users[50]. Likewise, when controls expected to receive MP, metabolism increased in the ventromedial prefrontal cortex whether they received MP or a placebo[51]. Together these data point to an intriguing possibility that expectation or sensitization following repeated drug exposure causes enhanced responsivity to drugs in the ventromedial prefrontal cortex. This tracks with preclinical studies suggesting that the infralimbic cortex (whose homologous region in humans includes ventromedial prefrontal cortex) is most active after repeated drug use and serves as an "off-switch" for extinction behavior; in contrast, prelimbic cortex (whose homologous region in humans includes dACC) is most active during the initial exposure to drugs and serves as an "on-switch" for drug-seeking[52].

The opposite pattern of responses in dACC and insula (BOLD increases) and ventromedial prefrontal cortex (BOLD decreases) are consistent with our hypothesis that IV MP but not oral MP would trigger stimulation of D1R with a consequent increase in activation whereas both IV and oral MP would stimulate D2R leading to decreased activation. A question that remains unresolved is the mechanism underlying the greater sensitivity of the salience network to D1R stimulation and that of the ventromedial prefrontal cortex to D2R stimulation.

In our study, striatal BOLD did not significantly correlate with dopamine dynamics, in contrast to our predictions based on a prior model[34]. Null findings may reflect that energetic effects of stimulation are associated mostly with terminal projections and not the location of cell bodies[53]. Therefore, while dopamine D1R and D2R would result in activation and deactivation of striatal MSN neurons respectively, the BOLD responses would be observed downstream in projection regions in the direct (D1R) or the indirect (D2R) striato-thalamo-cortical pathway. Further, MP's effects reflect dopamine and norepinephrine transporter blockade with a consequent increase in dopamine and norepinephrine; and whereas only the dopamine terminals to the striatum express dopamine transporters, terminals to the frontal cortex express norepinephrine transporters[54]. As such, downstream regions like dACC, which receive convergent projections from dopamine-rich (e.g., striatum) and norepinephrine-rich (e.g., thalamus) structures may be most relevant for the conscious experience of drug reward. In addition, our hypothesis represented a simplified model that focused primarily on postsynaptic receptor stimulation. In general decreases in striatal D2R binding with [¹¹C]raclopride are interpreted to reflect predominantly postsynaptic D2R receptors since

their relative concentration in striatum is estimated to be 4 times higher than that of D2 autoreceptors[55,56]. Yet dopamine increases to MP are also likely to evoke presynaptic effects on D2 autoreceptors[57], which would have attenuated peak dopamine increases by inhibiting dopamine cell firing and striatal dopamine release, ultimately attenuating striatal BOLD responses. Nonetheless, our null findings in striatum closely parallel a recent PET-fMRI study suggesting that dopamine signaling may more directly impact activation of frontal cortex, including dACC, and fronto-striatal connectivity rather than local striatal activation[58].

There are several considerations to note regarding the study population and design. First, this study was conducted in participants naïve to stimulant drugs. While the initial subjective experience to stimulant drugs has predictive value in the future development of substance use disorder[59], whether the circuit responses identified here generalize to some or all forms of substance use disorders needs to be further investigated. Second, individuals were administered MP by clinical staff in a laboratory environment, which, while tightly controlled, has low ecological validity. A rich literature in rats[60–63] and humans[64] finds that the environmental context during drug exposure has a critical impact on subsequent drug use behavior. For instance, adult males drank significantly more alcohol when they were randomized to exposure in a simulated bar environment relative to a neutral laboratory setting[65]. PET studies have also demonstrated that personalized cocaine cues elicit dopamine increases in people who misuse cocaine, especially among those with high levels of craving[66–70]. Finally, while our behavioral paradigm included 'high' ratings as a marker of drug reward, the construct of reward involves much more than the subjective pleasurable effects of drugs. Genetic and epigenetic vulnerability, prior conditioning, availability of alternative rewarding stimuli, and baseline physical/emotional state (e.g., withdrawal vs. satiation) all impact the experience of drug reward and deserve further study in this context[71].

Notably, our study identified two distinct circuits similar to the pattern of brain lesions leading to clinical remission of addiction[28]. Patients who suffered stroke lesions to brain regions that had positive functional connectivity with dACC and insula (where we observed activation with fast dopamine increases), and lesions to brain regions that had negative functional connectivity with ventromedial prefrontal cortex (where we observed deactivation both with slow and fast dopamine increases) led to remission. Therefore, both studies support interventions to inhibit the ACC and insula and interventions to stimulate the ventromedial prefrontal cortex as strategies for the treatment of substance use disorder. Indeed, the dACC is being tested as a neuromodulation target to combat compulsive drug use[72] with preliminary findings showing decreases in cocaine self-administration[73], cue-induced alcohol craving[74], and heavy drinking days[75]. Critically, in the latter study, successful stimulation effects were associated with decreased connectivity between dACC and caudate. A key next step is to evaluate if inhibition of this circuit during drug administration blocks the subjective experience of drug reward, which could open new avenues to treat substance use disorders.

## Methods

Some of the PET data and the behavioral data ('high' ratings) were previously reported in a recent publication[38]. Here, we applied these data to investigate the relationship between dynamic dopamine changes and brain function assessed with fMRI and all of the primary fMRI results in this manuscript are novel and have not been published.

### Participants

Twenty healthy individuals ($36.1 \pm 9.6$ years old; 9 females) participated in the study (see Supplementary Table 1 for participant characteristics). Participants were recruited through referrals from the NIH Volunteer Office, the Patient Recruitment and Public Liaison (PRPL) Office, ResearchMatch.org, by word of mouth, and through

Institutional Review Board (IRB)-approved advertisements. All individuals provided informed consent to participate in this double-blind placebo-controlled study, which was approved by the IRB at the National Institutes of Health (Combined Neurosciences White Panel; Protocol 17-AA-0178). This study was registered at clinicaltrials.gov (trial NCT03326245; https://classic.clinicaltrials.gov/ct2/show/NCT03326245 on October 31, 2017). All participants self-reported no history of nicotine/tobacco use. All participants were compensated for study participation. Our sample is broadly representative of the Washington, D.C. metro area, but it is possible that self-selection bias is present, though it is unclear whether this would affect the primary outcomes of this trial.

Sample size was determined based on estimates from prior similar studies, since no consensus exists on expected effect sizes for the outcome measures in this trial. However, previous imaging studies that used pharmacological challenges observed large effect sizes for strong within-subject manipulations such as IV drug administration, including the imaging outcomes reported here. Even 20 mg oral MP (1/3 the oral dose here) affected measures of resting fMRI brain function with Cohen's D = 0.53 relative to placebo[76], and IV MP produced striatal dopamine increases with Cohen's D = 1.1 relative to placebo[77]. We used these studies as a guide, while accounting for some regression to the mean. Power analysis using G*Power software (version 3.1.9.4) determined that $n = 20$ would be needed to achieve a medium-to-large effect size (Cohen's $D = 0.65$) for a paired $t$-test with $\alpha = 0.05$, and $1-\beta = 0.8$.

### Exclusion criteria
Participants were screened to exclude major medical and neuropsychiatric disorders that can impact brain function (seizures, tics, agitation, anxiety, panic attacks, psychotic disorders, glaucoma, dementia), past or present history of substance use disorders (lack of drug use was confirmed with a urine drug screen for benzodiazepines, cocaine, methamphetamines, opiates and tetrahydrocannabinol on all scan days), heart abnormalities (confirmed with electrocardiography), hypertension requiring medication or arrhythmia, pregnancy (confirmed with a urine pregnancy test) or breastfeeding, medications that may interact with methylphenidate (stimulants, analgesics containing narcotics, anorexics, antianginal agents, antiarrhythmics, corticosteroids, antibiotics, anticholinergics, anticoagulants, anticonvulsants, antidepressants, antidiarrheal, antifungal, antihistamines, anti-hypertensives, anti-inflammatory; antineoplastics, antiobesity, anti-psychotics, antivirals, anxiolytics, hormones, insulin, lithium, muscle relaxants, psychotropic drugs, sedatives/hypnotics), or ferromagnetic body implants that are contraindicated for MRI.

### Experimental design
The procedure for the study is illustrated in Fig. 1. Each participant was scanned on 3 different days, 40 ± 35 days apart, under different pharmacological conditions: (1) oral-MP (60 mg) and iv-placebo (3 cc saline), (2) oral-placebo and IV-MP (0.25 mg/kg in 3 cc sterile water), and (3) oral-placebo and iv-placebo. The session order was randomized and blocked across every six participants. Participants and all research staff were blind to medication (MP or PL) or route of administration (oral or IV). The key to the session order was held by independent personnel at the NIH Clinical Center Pharmacy until trial completion. Data were collected at the NIH Clinical Center in Bethesda, Maryland from January 2018 to September 2021.

### PET/MRI acquisition
The participants underwent simultaneous PET/MRI imaging in a 3T Biograph mMR scanner (Siemens; Medical Solutions, Erlangen, Germany). All studies were initiated at noon to minimize circadian variability. Venous catheters were placed in the left dorsal hand vein for radiotracer injection, and in the right dorsal hand vein for intravenous injection of medications. Heart rate (HR), systolic and diastolic blood pressures (BPs) were continuously monitored throughout the study with an Expression MR400 patient monitor (Philips, Netherlands). Thirty minutes before tracer injection, either 60 mg of MP or placebo was administered orally (p.o.). The participant was then positioned in the scanner. Earplugs were used to minimize scanner noise and padding to minimize head motion. A T1-weighted dual-echo image was collected for attenuation correction using an ultrashort-TE (UTE) sequence (192 × 192 × 192 matrix, 1.56 mm isotropic resolution, TR = 11.94 ms, TE = 0.07 and 2.46 ms) for PET attenuation correction, and T1-weighted 3D magnetization-prepared gradient-echo (MPRAGE; TR/TI/TE = 2200/1000/4.25 ms; FA = 9°, 1 mm isotropic resolution) was used to map brain structure. List mode PET emission data were acquired continuously for 90 min and initiated immediately after a manual bolus injection of [11C]raclopride (dose = 15.7 ± 1.9 mCi; duration 5–10 s). Simultaneously, fMRI data were acquired continuously for 90 min with a single-shot echo planar imaging (EPI) sequence (TE/TR = 30/3000 ms, FOV = 192 × 192 mm, in-plane resolution = 3 × 3 mm, 1800 volumes, 36 slices/volume, slice thickness = 4 mm). Thirty minutes after [11C]raclopride injection, either 0.25 mg/kg MP or placebo was manually injected i.v. as a ~30-s bolus. The participants were instructed to stay as still as possible and keep their eyes open during the scan.

### High ratings
High rating prompts were displayed on a projector using a program (E-Prime Version 3.0) designed to minimize visual stimulation. A white cross was presented at central fixation on a black screen. Participants were instructed to stay awake, relax, look at the cross, and not think of anything in particular. Occasionally, the cross would turn into a number for 10 s, and participants responded to the question: "How high do you feel right now, on a scale of 1–10, with 1 being minimum and 10 being maximum?". The first number presented at the start of each scanning session was always 1, and subsequent presentations matched the participant's high rating from the prior time point. Participants used a button box in their right hand to record responses. A button pressed with the right middle finger moved the rating up, one digit at a time, whereas the other button pressed with the right index moved it down. High rating prompts occurred every 5 min from the onset of oral MP administration; then, at the onset of IV-MP administration, prompts occurred every minute for 20 min. This faster sampling was chosen to capture the fast changes in reward during the first 20 min after IV-MP administration[78]; then, prompts occurred every 5 min until the end of scanning.

### MRI preprocessing
The minimal preprocessing pipelines of the Human Connectome Project (HCP)[79] were used for image processing. Specifically, Free-Surfer 5.3.0 (http://surfer.nmr.mgh.harvard.edu) was used for automatic segmentation of anatomical MRI scans into cortical and subcortical gray matter ROIs[80]. Then, for the EPI images, the FSL Software Library (version 5.0; http://www.fmrib.ox.ac.uk/fsl)[81] was used for rigid body realignment, field map processing, co-registration to the anatomical T1 image, and spatial normalization to MNI space.

We further processed the EPI images for resting fMRI analysis, including: regression of white matter, CSF, and global signals using custom MATLAB code; and 5 mm full-width at half-maximum spatial smoothing, using FSL. For dynamic resting connectivity analysis only, we also bandpass filtered the fMRI data in the 0.01–0.1 Hz frequency range.

                                          

## PET image reconstruction

A 3-dimensional ordered-subset expectation-maximization (OSEM) algorithm[82] with 3 iterations, 21 subsets, an all-pass filter, 344 × 344 × 127 matrix, and a model of the point spread function of the system was used for PET image reconstruction. The reconstructed PET time series consisted of 48 time windows (30 frames of 1 min, followed by 12 frames of 2.5 min, and 6 frames of 5 min) each with 2.086-mm in-plane resolution and 2.032-mm slice thickness. Attenuation coefficients (μ-maps) estimated from the UTE data using a fully convolutional neural network[83] were used to correct for scattering and attenuation of the head, the MRI table, the gantry, and the radiofrequency coil. Standardized uptake values (SUVs) for [¹¹C]raclopride were calculated after normalization for body weight and injected dose, co-registered with the strictrual T1w map, and spatially normalized to MNI space using paramaters obtained from the HCP pipelines[79]. Relative SUV time series, SUVr($t$), were computed in MNI space by normalizing each SUV volume by its mean SUV in cerebellum, as defined in individual FreeSurfer segmentations.

## Statistical analysis

**Behavioral and cardiovascular responses to oral and IV MP.** We tested how behavioral (self-reported 'high') and cardiovascular (heart rate and systolic blood pressure) responses to oral MP (slow delivery), IV MP (fast delivery) and placebo differed in intensity and over time, using repeated measures drug condition × time ANOVA, with the *aov* function in R. The main effect of drug condition tested for differences in the overall change in intensity in these measures for each session, and the drug-condition × time interaction tested for differences in the temporal dynamics of these measures across the three sessions.

**PET image analysis: static assessment of dopamine receptor availability.** Time-activity curves were extracted for putamen, caudate, ventral striatum and cerebellum from SUV time series using individual FreeSurfer segmentations. The Logan Plot graphical analysis for reversible systems using the cerebellum as the reference tissue and equilibration time $t^* = 20$ min was used to map the distribution volume ratio (DVR) and non-displaceable binding potential (BPnd)[84], independently for each participant and session.

**PET image analysis: estimation of dynamic 'dopamine increases' to oral and IV MP.** Decades of clinical and preclinical research have demonstrated that [¹¹C]Raclopride is sensitive to synaptic dopamine concentration, as it has lower affinity for dopamine $D_2$-like receptors than endogenous dopamine[85–87]. Therefore, decreases in [¹¹C]Raclopride binding following administration of a dopamine-boosting drug like MP are a suitable proxy for 'dopamine increases'[88,89].

Several groups have further found that one can model the time course of [¹¹C]Raclopride binding to measure the temporal dynamics of dopamine receptor occupancy (and dynamics of dopamine increases in response to dopamine-boosting interventions such as MP). Some of the most popular methods include 'neurotransmitter PET' (ntPET)[90], the 'linear simplified reference region model' (LSSRM)[91], and the 'dynamic binding potential'[35].

Recently, we developed a similar approach that is optimally suited for the current experimental design[38] (for a demonstration of the similarities between this method and prior methods, and for advantages of the current method for this particular study design, see the following section). Briefly, we found that dynamic ΔSUVr changes between placebo and MP conditions parallel the dynamics of dopamine increases as a function of time induced by MP in the striatum, which can be characterized by a gamma cumulative distribution function. To estimate the average time-varying dopamine increases to MP in the putamen we adjusted the amplitude, $A$, and the shape, $s$, parameters of the gamma cumulative distribution function

$$F(t) = \frac{A}{\Gamma(s)} \int_0^t e^{-x} x^{s-1} dx, \tag{1}$$

to fit $F(t)$ to the average ΔSUVr($t$) data across the 20 participants with the Levenberg-Marquardt algorithm for non-linear least-squares fitting in the interactive data language (IDL, L3Harris Geospatial, Boulder, CO). The corresponding probability density function, $f(t) = dF(t)/dt$ was used to estimate the average rate of dopamine increases at 1-min temporal resolution, independently for oral- and IV-MP, and were used as the regressors of interest for the estimation of fMRI activation.

The SUVr method can be seen as an approximation of LSSRM (see demonstration below). LSSRM requires only one scan session with an MP challenge to estimate dynamic dopamine increases, but it necessitates five fit parameters, which hindered reliable quantification of dopamine in our data. This may be in part because we were unable to continuously infuse [¹¹C]raclopride throughout the 90 min of scanning (this was due to the challenges posed by the simultaneous PET-MRI setup. Specifically, to ensure safety, our magnetic pump had to be positioned six feet away from the MRI bore. Consequently, utilizing the bolus-plus-infusion method for [¹¹C]raclopride would have necessitated excessively high levels of initial radioactivity (>80 mCi), which was deemed unsafe.) Therefore, radioactivity counts were lower at the end of the scan than in a paradigm with a continuous infusion. While LSSRM does not strictly require a paradigm with a continuous infusion, in our dataset we found that the relatively low radioactivity counts made dopamine quantification with LSSRM challenging. However, our design had the advantage of an additional placebo scan for each participant. Therefore, we developed an approach that capitalized on the added reliability the placebo scan affords, and could overcome the lack of a continuous [¹¹C]raclopride infusion. While the ΔSUVr approach requires two scans (MP and placebo) it has an important advantage: it only requires the amplitude of ΔSUVr and the time-to-peak of its derivative for fitting the ΔSUVr data, which improved the reliability of dynamic 'dopamine increases' estimates over prior methods.

**Comparison of ΔSUVr method for estimating 'dynamic dopamine increases' with prior methods.** The Simplified Reference Tissue Model (SRTM) defines the kinetic $C_T(t)$ of a target region in relation to the kinetic $C_R(t)$ of a reference region[92].

$$C_T(t) = R_1 C_R(t) + k_2 \int_0^t C_R(u)du - k_{2a} \int_0^t C_T(u)du \tag{2}$$

$R_1 = K'_1/K_1$ represents the local rate of delivery in the target tissue compared to the reference tissue, with $k_2$ representing the transfer rate constant from tissue to blood in the reference region, and $k_{2a}$ representing the transfer rate constant from tissue to blood in the target region. The linear extension of the simplified reference region model (LSSRM)[91] extended this model by incorporating a time-varying efflux rate $k_{2a}(t) = k_{2a} + \gamma h(t)$ that accounts for the competition between the radioligand and the endogenous neurotransmitter at the receptor sites. Here $\gamma$ represents the magnitude of transient effects and the function $h(t)$ characterizes the endogenous neurotransmitter discharge or an exogenous concurrent drug concentration level. Since MP increases extracellular dopamine, it also increases binding competition and reduces tracer concentration in the target region, Eq. (1) can be expressed as:

$$C_T^{MP}(t) = R_1 C_R(t) + k_2 \int_0^t C_R(u)du - k_{2a} \int_0^t C_T(u)du - \gamma \int_0^t C_T(u)h(u)du \tag{3}$$

The standardized uptake value, SUVr($t$), is calculated by dividing the uptake value in a specific region of interest (ROI) by the uptake value in a reference region. The reference region is typically an area of the brain that is considered to have minimal specific binding for the radiotracer used in the PET study. The SUVr is used as a simplified way to quantify the relative accumulation or binding of a radiotracer in a particular brain region compared to the reference region.

$$\text{SUVr}(t) = \frac{C_T(t)}{C_R(t)} \quad (4)$$

The SUVr is beneficial because it allows for comparison and analysis of PET data across different individuals or studies by normalizing the values to a reference region. This normalization accounts for potential variations in overall radiotracer uptake due to factors such as individual differences in blood flow or metabolism.

The SUVr change, ΔSUVr($t$), caused by MP-related increases in endogenous dopamine quantifies the change in radiotracer binding with respect to the placebo condition.

$$\Delta\text{SUVr}(t) = \frac{C_T(t) - C_T^{MP}(t)}{C_R(t)} \quad (5)$$

Inserting (1) and (2) in (4) ΔSUVr($t$) can be expressed as

$$\Delta\text{SUVr}(t) = \frac{\gamma \int_0^t C_T(u)h(u)du}{C_R(t)} \quad (6)$$

The instantaneous tissue concentration in the reference region $C_R(t)$ is described by the operational equation of the one-tissue compartment model:

$$\frac{dC_R(t)}{dt} = K_1 C_p(t) - k_2 C_R(t), \quad (7)$$

where the uptake rate constant $K_1 = 0.092$ mL/min.g and $k_2 = 0.45$ min$^{-1}$ (see ref. [91]). The plasmatic input function can be represented by the tri-exponential function

$$C_p(t) = \begin{cases} \frac{(A_1 + A_2 + A_3)}{t_{peak}} t \; \text{if} \; t < t_{peak} \\ \sum_{i=1}^{3} A_i \exp\left(-\frac{\ln(2)}{T_i}\left(t - t_{peak}\right)\right) \; \text{if} \; t \geq t_{peak} \end{cases}, \quad (8)$$

with $\vec{A} = (A_1, A_2, A_3) = (288.6, 1.1, 409.7) Bq/ml$, $\vec{T} = (T_1, T_2, T_3) = (4.28, 735.5, 183.5)$ sec, and $t_{peak} = 110$ s (see ref. [93]). The concentration of the tracer in the striatum can be simulated using specific parameters ($R_1 = 1.154$, and $k_{2a} = 0.065$ min$^{-1}$)[30] and Eq. (1). Supplementary Fig. 10A shows that $C_R(t)$ peaks earlier than $C_T(t)$, which reaches a maximum near the MP injection time ($t = 30$ min). In addition, Eq. (6) can be approximated as:

$$\Delta SUVr(t) \propto \int_0^t h(u)du \quad (9)$$

where $h(t)$ was modeled by a gamma probability distribution function (Supplementary Fig. 10B)[94]. A high correlation ($r = 0.987$) between ΔSUVr($t$) and $\int_0^t h(u)du$ was obtained in a 60 min window centered at the time of MP injection (Supplementary Fig. 10C, D). This first order approximation shows that ΔSUVr($t$) is proportional to the accumulation of endogenous dopamine caused by MP, which in our approach is represented by $F(t)$, and that the instantaneous dopamine change $h(u)$ is equivalent to the rate of dopamine, which in our approach is represented by $f(t)$.

The ΔSUVr approach offers a significant advantage over prior methods, such as LSSRM, by eliminating the need for individual-specific SRTM parameters ($R_1$, $k_2$, $k_{2a}$) to estimate dopamine increases. This enhances the robustness of model fitting as it only requires the amplitude of ΔSUVr and the time-to-peak of its derivative for fitting the ΔSUVr data.

**fMRI image analysis: activity changes in response to slow and fast dopamine increases.** To identify how brain activity was associated with the differing pharmacokinetic patterns of dopamine increases to slow (oral) vs. fast (IV) MP, we performed whole-brain voxelwise multiple regression analysis of fMRI images in SPM. We used $f(t)$, the PET-derived estimates of the rate of dynamic dopamine increases to oral and IV MP (average of all 20 participants), as the regressors of interest. Because these estimates were of lower temporal resolution (1 min) than the fMRI images (3 s), we first upsampled the dynamic dopamine increases to match the number of fMRI volumes via interpolation, using Python's *interp1d* function in the Scipy package with the 'extrapolate' method for filling missing values. Then, for each individual and for each session (placebo, oral MP, and IV MP) we used the time course of the rate of oral and IV dopamine increases as regressors against the whole-brain voxelwise maps of BOLD signal intensity, and additionally included a monotonically increasing linear term as of no interest, to account for linear drift in the fMRI signal. This analysis yielded whole-brain maps showing where brain activity was significantly associated with dynamic dopamine increases across time. We then subjected these maps (i.e., the contrast values from the multiple regression) to second-level analysis in SPM: one-sample $t$-tests for each drug condition, and then paired $t$-tests identifying the effects of each MP dose (for slow dopamine increases, we compared the oral vs. placebo conditions, and for fast dopamine increases, we compared the IV vs. placebo dopamine increases). For paired $t$-test analyses, we included drug condition order (binary, placebo first or second) as a covariate of no interest. We also conducted an additional control analysis to ensure that findings were driven by the rate and not the magnitude of dopamine increases; here he conducted another multiple regression similar to the prior analysis, except we also included the amplitude of dopamine increases (estimated by PET) as an additional regressor (for those results see Supplementary Figs. 4–5). For all whole-brain group level analyses, here and in the following sections, the significance threshold was set at voxelwise $p < 0.001$ uncorrected, with a cluster-forming threshold of $p < 0.05$ false discovery rate (FDR)-corrected, and a minimum cluster size of k > 50, in line with current reporting guidelines.

**fMRI image analysis: connectivity changes in response to slow and fast dopamine increases.** We then tested whether brain regions whose activity significantly associated with slow (oral) or fast (IV) dopamine increases showed dynamic connectivity patterns in association with slow or fast dopamine increases. To do this, we took the significant clusters from the paired $t$-test activation analyses to slow (oral vs. placebo) or fast (IV vs. placebo) MP, and using them as seed regions, computed dynamic functional connectivity across the 90-min scanning session (the actual number of time-points of dynamic functional connectivity was 82, due to inability to estimate connectivity for the first and last 4 min of the session). Connectivity, i.e., the z-scored temporal correlation between the bandpass-filtered BOLD signal intensity of the seed region and that of every other voxel in the brain, was computed with a 5-min sliding window with a 4-min overlap, yielding a connectivity estimate for each minute of the scan. Then, as with the activity analysis, for each individual and for each session (placebo, oral MP, and IV MP) we used the time course of both the oral and IV dopamine increases as regressors against the whole-brain voxelwise dynamic connectivity maps. As in the activity analysis, we performed one-sample and paired $t$-tests on these maps using drug condition order as a covariate.

**Brain connectivity changes in association with individual ratings of drug 'high'.** Finally, we sought to understand individual differences in the neurocircuitry behind the subjective experience of drug reward. We tested the association between dynamic brain connectivity (using any circuits identified as significantly associated with speed of dopamine increases, from the prior dynamic functional connectivity analysis) and 'high' ratings for each individual. We took each individual's self-reported 'high' ratings over the 90-min scan and interpolated the values to match the number of timepoints of dynamic connectivity (82) with the same interpolation method we used for dopamine increases. We then performed time series regression analysis of dynamic functional connectivity with high ratings using the 'dyn' and 'lm' packages in R. Analysis could only be performed in individuals who rated some change in 'high' over the course of the scan (that is, rated more than 1 out of 10 for at least one time point). This left $n = 19$ for the fast (IV) MP session but only $n = 13$ for the slow (oral) session; therefore, this analysis was only performed for the IV session. Then, second-level analysis consisted of a one-sample $t$-test for the IV session (the paired $t$-test comparing IV vs. placebo could not be performed because only 7 of 20 individuals reported feeling any 'high' during the placebo session, and so a regression of brain activity/connectivity with 'high' could not be computed for most participants).

### Reporting summary

Further information on research design is available in the Nature Portfolio Reporting Summary linked to this article.

## Data availability

The deidentified summary data generated in this study have been deposited in the Open Science Framework (OSF) database and are publicly available at https://osf.io/c58bf/.[95] Deidentified individual level data is available upon request to the corresponding author. Source data are also included in the Source data file. Source data are provided with this paper.

## Code availability

Code to produce primary analyses have been deposited in the Open Science Framework (OSF) database and are publicly available at https://osf.io/c58bf/.[95]

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

## Acknowledgements
We would like to thank Minoo MacFarland and John Butman for their contributions to data acquisition and processing. This research was supported (in part) by the Intramural Research Program of the NIH (ZIAAA000550; National Institute on Alcohol Abuse and Alcoholism; P.M.).

## Author contributions
P.M., D.T., and N.D.V. designed the study and interpreted the data; P.M. and D.T. analyzed the data, developed code, and performed simulations; P.M., M.V.Y., D.K., D.F., K.M., C.B., E.D., A.J., W.T.W., and G.J.W. collected data; E.S.K. and R.Z. provided image preprocessing; K.Y. developed software; P.M. and N.D.V. wrote the manuscript with contributions from all co-authors.

## Funding

## Competing interests
The authors declare no competing interests.
