## [Peer Review File · Nature Communications]

Neural circuit selective for fast but not slow dopamine increases in drug rewardReviewers' Comments:

Reviewer #1 (Remarks to the Author):

This is a manuscript reporting on the results of an experiment using PET-fMRI to examine links between subjective ratings of 'high', tracer binding to dopamine D2/D3 receptors as an estimate of changes in extracellular dopamine, and BOLD fMRI signal intensity as an estimate of functional activity and connectivity in the brain, all following slow-onset (oral) or fast-onset (i.v.) methylphenidate, in women and men. The results are informative as regards the subjective and neurobiological effects of methylphenidate as a function of route/rate of administration. The authors report that a circuit including the dorsal anterior cingulate cortex and dorsal caudate is potentially linked to the 'high' ratings and dopamine response to i.v. but not oral methylphenidate.

The findings are new and noteworthy. I recommend the following revisions.

Abstract

- In the abstract the term 'to triangulate' is a distraction, please replace with 'to link/associate/explore', or something akin.
- 'These data provide first evidence in humans of the involvement of dACC activation uniquely by fast dopamine increases and of its key role in drug reward...'. The sentence is difficult to follow and should be revised. Specifically the authors should clarify what they mean by 'the involvement of dACC activation by fast dopamine increases'.

Introduction

- 'The rate at which dopamine increases along with the magnitude of the increase determine the intensity of a drug's rewarding effects and its addictive potential'. A review is cited to support this statement but original studies should be cited here as well, and these include PMID: 18396266, PMID: 15254092 and PMID: 29757478.
- 'As such, routes of drug administration that result in the fastest delivery into the brain (e.g., intravenous injection and smoking) are more rewarding than when taken orally (with doses adjusted to reach equivalent levels in plasma)²⁻⁴.' Is reference number 3 appropriate here? It appears to be a paper on the effects of intermittent alcohol intake.

- ‘Further, faster self-administration of IV alcohol is strongly associated with risk factors for alcohol use disorder³’ Similarly, is reference number 3 appropriate here? It does not report on i.v. alcohol administration. There is also a date mistakenly inserted after this sentence.

- ‘despite equivalent levels of dopamine transporter blockade by cocaine in the brain of cocaine users, the rewarding effects were significantly higher for intranasal and intravenous administration than for insufflation¹⁵’. However in the cited study, it was concluded that self-reports of high were greatest after smoked versus intranasal or intravenous cocaine.

- In interpreting the results of the study in reference 15, the authors suggest that this likely reflects and involvement of extra-striatal responses in the subjective experience of drug reward. However an equally likely, and non-mutually exclusive, possibility is that the rewarding effects of drugs like methylphenidate are linked to aspects of the dopaminergic response beyond maximum levels of dopamine transporter blockade. Specifically this could involve the rate of rise of dopamine transporter blockade and ensuing rate of accumulation of dopamine in the extracellular space, and resulting dopamine receptor stimulation, as suggested by the following series of studies; PMID: 18396266, PMID: 15254092 and PMID: 29757478.

Results

- Figures 2-3 illustrate data starting at time since oral drug administration. Oral and i.v. dosing of methylphenidate did not occur on the same scanning sessions, so unless there's a convincing reason not to do so, the data should be aligned across routes of administration, and illustrated starting from ‘time 0’ for each route of administration, where time 0 is the moment when the participants received the oral or i.v. dose.

- Extracellular dopamine concentrations were not measured directly, but inferred from changes in binding of [¹¹C]raclopride. This should be highlighted more explicitly in the text, and where appropriate, the term ‘dopamine increases’ should be replaced with the actual dependent measure, which is linked to a change in tracer binding to dopamine D2/D3 receptors.

- Lines 119-120; ‘which we used for subsequent analyses to identify brain circuits that synergized with dopamine dynamics’. This sentence should be rephrased to state ‘...which we used for subsequent analyses to identify brain circuits were activity coincided in time with dopamine dynamics’, if that is indeed what was done here. And the term ‘dopamine dynamics’ here should be revised to indicate the actual dependent variable used in these analyses (presumably a change in dopamine D2/D3 receptor binding).

- Supplementary figures 1, 3 should have a legend identifying the different colour curves

Discussion

- First sentence 'Together, these findings provide insight for why..' Please correct to 'Together, these findings provide insight into..'
- In the introduction or discussion the authors should address the addictive potential of i.v. methylphenidate. Are there statistics available on use, prevalence and effects of this substance i.v.? This should include discussion of any data that would support the idea that i.v. use of methylphenidate increases the potential for addiction to the drug. If no such convincing data exist, the authors should acknowledge then that methylphenidate was used as a prototypical drug, as proof of concept.
- In the first paragraph of the discussion, the statement '...downstream cortical targets of the caudate appear necessary for the conscious experience of drug reward' needs to be clarified. As written, it is not clear if this is referring to cortical, monosynaptic inputs to the caudate or connections going from the caudate to cortical areas, which would not be monosynaptic and would involve additional brain regions (e.g., globus pallidus and thalamus).
- The authors should also address the fact that participants were naive to stimulant drugs, and to the best of my understanding, they were also exposed to methylphenidate only twice. While I understand ethical concerns underlying this experimental design, and the design does provide informative results, I question drawing conclusions about both substance use disorder and its treatment based on data obtained after an acute administration of methylphenidate, to people who do not have a history of chronic drug use. This should be acknowledged as a consideration in interpreting the results and a limit to generalizing these results to substance use disorder. As an example, the authors cite a study showing that after a single dose of methylphenidate i.v., metabolism in frontal cortex can decrease whereas it can increase after repeated doses.
- Another critical point to discuss is that a very rich literature in both humans and other laboratory animals indicates that when drugs are given in environmental contexts that have not previously been paired with drug use or drug effects (i.e., that effectively signal the absence of drug and drug effects), as was done here, this can change the neurobiological and behavioral response to the drug.
- Prior published work has used different methods to compare neuronal activity induced by fast-versus slow-onset cocaine, namely the following references; PMID: 7871040; PMID: 15254092; PMID: 18396266. This work should be cited and discussed.
- The text seems to equate subjective ratings of being high with drug reward. Drug reward can certainly include the subjective pleasurable effects of drugs, but the text should acknowledge that it is also much more than this.

A. Samaha

Reviewer #2 (Remarks to the Author):

The manuscript by Manza and colleagues describes a study using multi-modal techniques (PET and functional MRI) to study the circuits underlying the speed and magnitude of DA release in response to a fast (IV) vs. slow (PO) methylphenidate administration. This is a well-written paper, and the study design is rigorous. However, there are questions about the methods used and the data presented.

Overall, slow dopamine release via PET data was associated with decreased vmPFC activation and fast dopamine release from PET data was associated with decreased vmPFC activation and increased activation in a cluster including dorsal and middle anterior cingulate cortex and left insula. Inclusion of the subjects individual fmri data in Figure 2 is a strength. These clusters were then used as seed regions to perform functional connectivity - dACC-dorsal caudate connectivity was associated with fast dopamine release and with the 'high' ratings time course in the fast dopamine release data. The data demonstrate that the salience network is preferentially activated when highly reinforcing drugs with fast dopamine release are administered. It will be interesting to see how this paradigm and the findings translate to people with substance use disorders.

There have been methods available to examine transient DA changes (i.e., non-static PET analysis) since ~ 2003. And to do so with fully quantitative outcome measures. SUV is semi-quantitative. It's curious that a well characterized, validated, and quantitative method like LSRRM or ntPET or similar was not used. Why was the SUVr method chosen?

The PET data appears to be exactly the same as in the previous Tomasi 2023 paper – ref 30. At a minimum this should be clearly stated. Figure 3D is the same data as Fig 7A in the previous paper. All 3 conditions of PET data and the data about 'high' have been previously published as part of the methods development paper. The fMRI data and analyses are new.

It is misleading to reference a paper to justify the PET methods used and then to use that same data in the current paper.

For inclusion/exclusion the authors did a commendable job excluding for other substance use and comorbidity, however nicotine/tobacco smoking was not mentioned even though use of nicotine has published effects on the DA system.

Reviewer #3 (Remarks to the Author):

In this study, the researchers utilized a combined [11C]raclopride-PET/BOLD-fMRI technique to investigate the dynamics of dopamine release in the striatum and its correlation with overall brain activity and connectivity. They administered methylphenidate through both oral and intravenous routes to achieve slow and rapid delivery to the brain. The findings showed that rapid dopamine increases triggered activation in the frontostriatal circuit, which includes the dorsal anterior cingulate cortex (dACC) and dorsal caudate. However, oral administration did not result in any activation in this circuit. Interestingly, the subjects reported high ratings corresponding to the observed effects which leads to the conclusion that dACC activation plays a key role in drug reward and may be a promising target for neuromodulations to treat substance use disorders.

Figure 2 shows [11C]raclopride PET data on a temporal scale, which provides information about the rate and amplitude of DA (D2 binding decrease) in the striatum. This analysis approach does not allow for the examination of temporal fluctuations of DA in extrastriatal regions. Thanarajah et al. (<https://pubmed.ncbi.nlm.nih.gov/30595479/>) recently related BOLD activations to temporal changes in extrasynaptic DA release in several brain regions using a [11C]raclopride bolus infusion approach, and I believe this information would be very valuable for better interpreting the fMRI findings in the dACC.

The present study demonstrates nonetheless a well-designed methodology and data analysis. The results section could benefit from additional analysis and a more comprehensive discussion of the findings.

Comments:

1. The hypothesis put forward by the authors in Figure 1c proposes an explanation for the fMRI signal changes based on the opposing effects of D2 and D1 receptors in the indirect and direct pathways. However, it is my belief that this hypothesis may not comprehensively elucidate the observed alterations fMRI and functional connectivity (FC) following pharmacological stimulations that enhances synaptic dopamine levels. Apart from the effects on postsynaptic receptors, there may also be presynaptic effects on D2 autoreceptors that inhibit phasic neuronal firing and dopamine release, leading to attenuated neuronal signaling. This attenuation could potentially account for the lack of response in the striatum.

2. The authors state in the discussion section 'striatal BOLD did not significantly correlate with dopamine dynamics in this study', since they observed no BOLD changes in the striatum. Did the authors also check the caudate nucleus and putamen? At least from supplemental figure S4, bottom (amplitude of dopamine), there seems to be decreased activity related to putamen in the placebo and iv condition, while the oral administration shows increased activity in the Caudate. In S7 (insular seeded FC), there seems to be decreased connectivity in the oral group between insular and putamen? Please provide any significant clusters of the BOLD signal changes in a table.

3. Did the authors look at sex differences in their findings. Where the observed effects independent of the sex? I think this would be important to report even if there are no differences.

4. I would appreciate a more comprehensive explanation/discussion as to why the authors attribute the response to the intravenous drug administration to the dorsal (cognitive) region of the anterior cingulate cortex (ACC) rather than the rostral (affective) region. The ACC plays a significant role in various functions such as contextual memory, emotional regulation/appraisal, social cognition, resting-state default mode, and conflict monitoring. In the context of drug reward, what specific role does the ACC fulfill?

Minor comments:

1. Some groups have reported signal changes of [11C]raclopride-PET data in the cerebellum after pharmacological challenges. Could the authors provide the mean TAC files in the supplemental notes?

2. In figure 2f, there is a BOLD signal increase of the placebo group after treatment. This is not discussed in the discussion or results.

We thank the reviewers for their helpful comments. Our responses are listed below.

Reviewer #1 (Remarks to the Author):

This is a manuscript reporting on the results of an experiment using PET-fMRI to examine links between subjective ratings of 'high', tracer binding to dopamine D2/D3 receptors as an estimate of changes in extracellular dopamine, and BOLD fMRI signal intensity as an estimate of functional activity and connectivity in the brain, all following slow-onset (oral) or fast-onset (i.v.) methylphenidate, in women and men. The results are informative as regards the subjective and neurobiological effects of methylphenidate as a function of route/rate of administration. The authors report that a circuit including the dorsal anterior cingulate cortex and dorsal caudate is potentially linked to the 'high' ratings and dopamine response to i.v. but not oral methylphenidate.

The findings are new and noteworthy. I recommend the following revisions.

Abstract

- In the abstract the term 'to triangulate' is a distraction, please replace with 'to link/associate/explore', or something akin.
- 'These data provide first evidence in humans of the involvement of dACC activation uniquely by fast dopamine increases and of its key role in drug reward...'. The sentence is difficult to follow and should be revised. Specifically the authors should clarify what they mean by 'the involvement of dACC activation by fast dopamine increases'.

Response: We have made these revisions, and the conclusion in the abstract now reads:

These data provide first evidence in humans for a link between dACC activation and fast but not slow dopamine increases. They also demonstrate dACC's key role in drug reward, revealing a promising target for neuromodulation interventions to treat substance use disorders.

Introduction

- 'The rate at which dopamine increases along with the magnitude of the increase determine the intensity of a drug's rewarding effects and its addictive potential'. A review is cited to support this statement but original studies should be cited here as well, and these include PMID: 18396266, PMID: 15254092 and PMID: 29757478.

Response: Thank you for these suggestions; we have added these references.

- 'As such, routes of drug administration that result in the fastest delivery into the brain (e.g., intravenous injection and smoking) are more rewarding than when taken orally (with doses adjusted to reach equivalent levels in plasma)^{2–4}. Is reference number 3 appropriate here? It appears to be a paper on the effects of intermittent alcohol intake.

Response: We have removed reference 3.

- 'Further, faster self-administration of IV alcohol is strongly associated with risk factors for alcohol use

disorder³ Similarly, is reference number 3 appropriate here? It does not report on i.v. alcohol administration. There is also a date mistakenly inserted after this sentence.

Response: We apologize for this error, which appears to have happened during the PDF conversion on the submission site. We have changed this sentence to read:

Further, the fastest routes of administration evoked the strongest self-reported pleasurable effects to IV cocaine (e.g., ‘high’, ‘liking’) (Resnick et al., 1977; Abreu et al., 2001).

And we will ensure this is properly encoded on resubmission.

- ‘despite equivalent levels of dopamine transporter blockade by cocaine in the brain of cocaine users, the rewarding effects were significantly higher for intranasal and intravenous administration than for insufflation¹⁵. However in the cited study, it was concluded that self-reports of high were greatest after smoked versus intranasal or intravenous cocaine.

Response: We have revised this to state:

“the rewarding effects were dependent on speed of delivery, with greatest ‘high’ reported for smoked cocaine (time to peak effects: 1.4 min), followed by intravenous (3.1 min) and then intranasal (14.6 min) routes of administration”

- In interpreting the results of the study in reference 15, the authors suggest that this likely reflects and involvement of extra-striatal responses in the subjective experience of drug reward. However an equally likely, and non-mutually exclusive, possibility is that the rewarding effects of drugs like methylphenidate are linked to aspects of the dopaminergic response beyond maximum levels of dopamine transporter blockade. Specifically this could involve the rate of rise of dopamine transporter blockade and ensuing rate of accumulation of dopamine in the extracellular space, and resulting dopamine receptor stimulation, as suggested by the following series of studies; PMID: 18396266, PMID: 15254092 and PMID: 29757478.

Response: We agree with this possibility and do not see it as being in conflict with a role for extra-striatal involvement in drug reward. In paragraph 1 of the introduction, we had cited the faster rise of striatal dopamine increases as one of the mechanisms associated with the unique responses to fast brain delivery of cocaine. A fast rise in striatal dopamine is a key part of the rewarding properties of drugs delivered quickly, and still compatible with the interpretation that, *“the conscious experience of drug reward depends on more than local striatal responses, since dopamine signaling activates large-scale downstream networks via reciprocal cortical connections”*

Especially given the robust and reproducible finding in humans that lesions to specific areas of neocortex can abolish the rewarding properties of drugs.

The evidence from the studies noted by this reviewer, as well as prior studies in humans, are precisely why we have used the rate of dopamine increases as the key measure for understanding whole-brain fMRI responses to methylphenidate, and why a significant proportion of the discussion is devoted to how the rate of dopamine increases affects brain and behavioral responsivity to drugs.

Results

- Figures 2-3 illustrate data starting at time since oral drug administration. Oral and i.v. dosing of methylphenidate did not occur on the same scanning sessions, so unless there's a convincing reason not to do so, the data should be aligned across routes of administration, and illustrated starting from 'time 0' for each route of administration, where time 0 is the moment when the participants received the oral or i.v. dose.

Response: We appreciate this suggestion. However, participants did in fact receive both an oral dose and an IV dose in each session (i.e., in the 'Oral' Session, they also received an IV placebo dose, and in the 'IV' session, they also received an oral placebo dose). We feel it's important to show the entire data time series to demonstrate that there are not significant placebo effects in response to these different administrations.

- Extracellular dopamine concentrations were not measured directly, but inferred from changes in binding of [¹¹C]raclopride. This should be highlighted more explicitly in the text, and where appropriate, the term 'dopamine increases' should be replaced with the actual dependent measure, which is linked to a change in tracer binding to dopamine D2/D3 receptors.
- Lines 119-120; 'which we used for subsequent analyses to identify brain circuits that synergized with dopamine dynamics'. This sentence should be rephrased to state '...which we used for subsequent analyses to identify brain circuits were activity coincided in time with dopamine dynamics', if that is indeed what was done here. And the term 'dopamine dynamics' here should be revised to indicate the actual dependent variable used in these analyses (presumably a change in dopamine D2/D3 receptor binding).

Response: PET studies using tracers like [¹¹C]Raclopride, which exhibit competitive binding to dopamine receptors, have been utilized for over 30 years. Convergent evidence from decades of clinical and preclinical study suggests it is not controversial to claim that changes in [¹¹C]Raclopride binding following administration of methylphenidate (or other stimulant drugs) are a suitable proxy for 'dopamine increases'. Indeed, it is convention in this field to use this terminology.

Nevertheless, we agree that it's important to state the dependent measure more explicitly. In the manuscript, we now write:

Conventional 'static' analysis of PET imaging revealed significant decreases in relative standardized uptake value (SUVr) to oral (slow) and IV (fast) MP; these changes in [¹¹C]raclopride's specific binding to the drug administrations are a widely accepted measure of increases in synaptic dopamine concentrations (**Figure 2a**). Henceforth we refer to these drug-induced decreases in [¹¹C]raclopride's specific binding as 'dopamine increases'.

We also now describe in the methods section some of the foundational studies that demonstrate the relationship between MP-induced decreases in [¹¹C]Raclopride binding in striatum and dopamine increases (methods p. 14).

We have also replaced the term 'synergized' with the suggested phrasing.

- Supplementary figures 1, 3 should have a legend identifying the different colour curves

Response: We have added these legends.

Discussion

- First sentence 'Together, these findings provide insight for why..' Please correct to 'Together, these findings provide insight into..'

Response: We have made this change.

- In the introduction or discussion the authors should address the addictive potential of i.v. methylphenidate. Are there statistics available on use, prevalence and effects of this substance i.v.? This should include discussion of any data that would support the idea that i.v. use of methylphenidate increases the potential for addiction to the drug. If no such convincing data exist, the authors should acknowledge then that methylphenidate was used as a prototypical drug, as proof of concept.

Response: We have added the following as the 2nd paragraph of the introduction:

Preclinical studies have long used stimulants including methylphenidate (MP) as model drugs to study the relationship between pharmacokinetics and drug reward. But MP also has clinical relevance for it is widely used to treat attention deficit hyperactivity disorder (ADHD). Because of its addictive potential (Morton and Stock, 2000) and clear evidence of misuse particularly when injected (Bjarnadottir et al., 2015), MP is classified by the FDA as a schedule II drug alongside other addictive drugs like cocaine and methamphetamine. Despite MP's addictiveness it can be used safely and it is therapeutic when given orally. This is why MP formulations have been developed that make it harder to inject or snort MP (i.e., tamper-resistant formulations) (Cortese et al., 2017).

- In the first paragraph of the discussion, the statement '...downstream cortical targets of the caudate appear necessary for the conscious experience of drug reward' needs to be clarified. As written, it is not clear if this is referring to cortical, monosynaptic inputs to the caudate or connections going from the caudate to cortical areas, which would not be monosynaptic and would involve additional brain regions (e.g., globus pallidus and thalamus).

Response: Both types of connections are potentially involved, and we mentioned in the introduction that communication between these regions supporting drug reward may be mediated "via reciprocal cortical connections". However, with fMRI we are unable to resolve whether one or both types of inputs are definitively contributing to the findings here. We now also state in the discussion:

"This pattern may be driven by monosynaptic inputs from cortex to caudate, and/or by polysynaptic connections from caudate to cortex via pallidum and thalamus; future preclinical studies could resolve the circuit mechanisms in greater detail."

- The authors should also address the fact that participants were naive to stimulant drugs, and to the best of my understanding, they were also exposed to methylphenidate only twice. While I understand ethical concerns underlying this experimental design, and the design does provide informative results, I question drawing conclusions about both substance use disorder and its treatment based on data obtained after an acute administration of methylphenidate, to people who do not have a history of

chronic drug use. This should be acknowledged as a consideration in interpreting the results and a limit to generalizing these results to substance use disorder. As an example, the authors cite a study showing that after a single dose of methylphenidate i.v., metabolism in frontal cortex can decrease whereas it can increase after repeated doses.

- Another critical point to discuss is that a very rich literature in both humans and other laboratory animals indicates that when drugs are given in environmental contexts that have not previously been paired with drug use or drug effects (i.e., that effectively signal the absence of drug and drug effects), as was done here, this can change the neurobiological and behavioral response to the drug.
- The text seems to equate subjective ratings of being high with drug reward. Drug reward can certainly include the subjective pleasurable effects of drugs, but the text should acknowledge that it is also much more than this.

Response: We agree that all of these points are important considerations. We have added the following to the discussion p. 10:

There are several considerations to note regarding the study population and design. First, this study was conducted in participants naïve to stimulant drugs. While the initial subjective experience to stimulant drugs has predictive value in the future development of substance use disorder (Lambert et al., 2006), whether the circuit responses identified here generalize to some or all forms of substance use disorders needs to be further investigated. Second, individuals were administered MP by clinical staff in a laboratory environment, which, while tightly controlled, has low ecological validity. A rich literature in rats (Crombag et al., 2008) and humans (LeCocq et al., 2020) finds that the environmental context during drug exposure has a critical impact on subsequent drug use behavior. For instance, adult males drank significantly more alcohol when they were randomized to exposure in a simulated bar environment relative to a neutral laboratory setting (Lau-Barraco and Dunn, 2009). Finally, while our behavioral paradigm included ‘high’ ratings as a marker of drug reward, the construct of reward involves much more than the subjective pleasurable effects of drugs. Genetic and epigenetic vulnerability, prior conditioning, availability of alternative rewarding stimuli, and baseline physical/emotional state (e.g., withdrawal vs. satiation) all impact the experience of drug reward and deserve further study in this context (Volkow et al., 2019).

- Prior published work has used different methods to compare neuronal activity induced by fast- versus slow-onset cocaine, namely the following references; PMID: 7871040; PMID: 15254092; PMID: 18396266. This work should be cited and discussed.

Response: Thank you for these references. We had previously cited Samaha et al. 2004 and now additionally cite Porrino 1993 and Ferrario et al. 2008 in our introduction, p.3, in the context of how fast versus slow-onset cocaine administration is associated with faster rise in striatal dopamine, greater metabolic activity in brain reward circuitry, and more cocaine self-administration.

Reviewer #2 (Remarks to the Author):

The manuscript by Manza and colleagues describes a study using multi-modal techniques (PET and functional MRI) to study the circuits underlying the speed and magnitude of DA release in response to a fast (IV) vs. slow (PO) methylphenidate administration. This is a well-written paper, and the study design is rigorous. However, there are questions about the methods used and the data presented.

Overall, slow dopamine release via PET data was associated with decreased vmPFC activation and fast dopamine release from PET data was associated with decreased vmPFC activation and increased activation in a cluster including dorsal and middle anterior cingulate cortex and left insula. Inclusion of the subjects individual fmri data in Figure 2 is a strength. These clusters were then used as seed regions to perform functional connectivity - dACC-dorsal caudate connectivity was associated with fast dopamine release and with the 'high' ratings time course in the fast dopamine release data. The data demonstrate that the salience network is preferentially activated when highly reinforcing drugs with fast dopamine release are administered. It will be interesting to see how this paradigm and the findings translate to people with substance use disorders.

There have been methods available to examine transient DA changes (i.e., non-static PET analysis) since ~ 2003. And to do so with fully quantitative outcome measures. SUV is semi-quantitative. It's curious that a well characterized, validated, and quantitative method like LSSRM or ntPET or similar was not used. Why was the SUVr method chosen?

Response: This is an important point. The SUVr method is essentially an approximation of LSSRM/ntPET methods, which utilize a combination of bolus injection and continuous infusion of the radiotracer to maintain a stable concentration in blood. The continuous infusion is achieved using a programmable pump, which controls the infusion rate and duration. We attempted the bolus+continuous infusion approach when first designing the study, but safety concerns related to the magnetic environment of the MRI scanner required that the pump be placed 6 feet away from the magnet, making continuous administration of the radiotracer unfeasible (it required higher radioactivity levels and greater exposure of personnel). Due to these safety concerns, we developed the SUVr method as a suitable and highly similar alternative.

We now include simulations that demonstrate the extremely high correspondence between the SUVr and LSSRM methods. For full details please see the following in green (now listed in the methods pp. 14-17).

In brief, the SUVr method can be seen as an approximation of LSSRM (see demonstration below). LSSRM requires only one scan session with an MP challenge to estimate dynamic dopamine increases, but it necessitates five fit parameters, which hindered reliable quantification of dopamine in our data. This may be in part because we were unable to continuously infuse

[¹¹C]raclopride throughout the 90 minutes of scanning¹, and therefore the radioactivity counts were lower at the end of the scan than in a paradigm with a continuous infusion. However, our design had the advantage of an additional placebo scan for each participant. Therefore, we developed an approach that capitalized on the added reliability the placebo scan affords, and could overcome the lack of a continuous [¹¹C]raclopride infusion. While the ΔSUVr approach requires two scans (MP and placebo) it has an important advantage: it only requires the amplitude of ΔSUVr and the time-to-peak of its derivative for fitting the ΔSUVr data, which improved the reliability of dynamic ‘dopamine increases’ estimates over prior methods.

Comparison of ΔSUVr method for estimating ‘dynamic dopamine increases’ with prior methods.

The Simplified Reference Tissue Model (SRTM) defines the kinetic $C_T(t)$ of a target region in relation to the kinetic $C_R(t)$ of a reference region (Lammertsma and Hume, 1996).

$$C_T(t) = R_1 C_R(t) + k_2 \int_0^t C_R(u) du - k_{2a} \int_0^t C_T(u) du \quad [1]$$

$R_1 = K'_1/K_1$ represents the local rate of delivery in the target tissue compared to the reference tissue, with k_2 representing the transfer rate constant from tissue to blood in the reference region, and k_{2a} representing the transfer rate constant from tissue to blood in the target region. The linear extension of the simplified reference region model (LSSRM; (Alpert et al., 2003)) extended this model by incorporating a time-varying efflux rate $k_{2a}(t) = k_{2a} + \gamma h(t)$ that accounts for the competition between the radioligand and the endogenous neurotransmitter at the receptor sites. Here γ represents the magnitude of transient effects and the function $h(t)$ characterizes the endogenous neurotransmitter discharge or an exogenous concurrent drug concentration level. Since MP increases extracellular dopamine, it also increases binding competition and reduces tracer concentration in the target region, Eq [1] can be expressed as:

$$C_T^{MP}(t) = R_1 C_R(t) + k_2 \int_0^t C_R(u) du - k_{2a} \int_0^t C_T(u) du - \gamma \int_0^t C_T(u) h(u) du \quad [2]$$

The standardized uptake value, $SUVr(t)$, is calculated by dividing the uptake value in a specific region of interest (ROI) by the uptake value in a reference region. The reference region is typically an area of the brain that is considered to have minimal specific binding for the radiotracer used in the PET study. The $SUVr$ is used as a simplified way to quantify the relative

¹ We were unable to do a continuous [¹¹C]raclopride infusion due to the challenges posed by the simultaneous PET-MRI setup. More specifically, to ensure safety, our magnetic pump had to be positioned six feet away from the MRI bore. Consequently, utilizing the bolus-plus-infusion method for [¹¹C]raclopride would have necessitated excessively high levels of initial radioactivity (>80mCi), which was deemed unsafe.

accumulation or binding of a radiotracer in a particular brain region compared to the reference region.

$$SUVr(t) = \frac{C_T(t)}{C_R(t)} \quad [3]$$

The *SUVr* is beneficial because it allows for comparison and analysis of PET data across different individuals or studies by normalizing the values to a reference region. This normalization accounts for potential variations in overall radiotracer uptake due to factors such as individual differences in blood flow or metabolism.

The *SUVr* change, $\Delta SUVr(t)$, caused by MP-related increases in endogenous dopamine quantifies the change in radiotracer binding with respect to the placebo condition.

$$\Delta SUVr(t) = \frac{C_T(t) - C_T^{MP}(t)}{C_R(t)} \quad [4]$$

Inserting [1] and [2] in [4] $\Delta SUVr(t)$ can be expressed as

$$\Delta SUVr(t) = \frac{\gamma \int_0^t C_T(u) h(u) du}{C_R(t)} \quad [5]$$

The instantaneous tissue concentration in the reference region $C_R(t)$ is described by the operational equation of the one-tissue compartment model:

$$\frac{dC_R(t)}{dt} = K_1 C_p(t) - k_2 C_R(t), \quad [6]$$

where the uptake rate constant $K_1 = 0.092$ mL/min.g and $k_2 = 0.45$ min⁻¹; see (Alpert et al., 2003). The plasmatic input function can be represented by the tri-exponential function

$$C_p(t) = \begin{cases} \frac{(A_1 + A_2 + A_3)}{t_{peak}} t & \text{if } t < t_{peak} \\ \sum_{i=1}^3 A_i \exp\left(-\frac{\ln(2)}{T_i} (t - t_{peak})\right) & \text{if } t \geq t_{peak} \end{cases}, \quad [7]$$

with $\vec{A} = (A_1, A_2, A_3) = (288.6, 1.1, 409.7) Bq/ml$, $\vec{T} = (T_1, T_2, T_3) = (4.28, 735.5, 183.5) sec$, and $t_{peak} = 110$ sec; see (Irace et al., 2020). The concentration of the tracer in the striatum can be simulated using specific parameters ($R_1 = 1.154$, and $k_{2a} = 0.065$ min⁻¹)³⁰ and Eq [1]. **Fig 1A** shows that $C_R(t)$ peaks earlier than $C_T(t)$, which reaches a maximum near the MP injection time ($t = 30$ min). Additionally, Eq [5] can be approximated as:

$$\Delta SUVr(t) \propto \int_0^t h(u) du$$

where $h(t)$ was modelled by a gamma probability distribution function (**Fig 1B**) (Normandin et al., 2012).

A high correlation ($r = 0.987$) between $\Delta SUVr(t)$ and $\int_0^t h(u) du$ was obtained in a 60 min window centered at the time of MP injection (**Figs 1C and 1D**). This first order approximation shows that $\Delta SUVr(t)$ is proportional to the accumulation of endogenous dopamine caused by MP, which in our approach is represented by $F(t)$, and that the instantaneous dopamine change $h(u)$ is equivalent to the rate of dopamine, which in our approach is represented by $f(t)$.

Figure 1: A) Time-varying concentrations of [11C]raclopride in the striatum, $C_T(t)$, and in the cerebellum, $C_R(t)$. B) A gamma variate function modeling the endogenous dopamine increases elicited by methylphenidate (MP), $h(t)$. C) Dynamics of SUVr changes, $\Delta SUVr(t)$, caused by MP-related increases in endogenous dopamine modeled with the LSSRM (black) and the approximation used in this study. D) Linear association between the exact and approximated LSSRM solutions. Normal random noise (5%) was added to $C_R(t)$ and $C_T(t)$. Dashed lines indicate the time of MP injection.

The $\Delta SUVr$ approach offers a significant advantage over prior methods, such as LSSRM, by eliminating the need for individual-specific SRTM parameters (R_1 , k_2 , k_{2a}) to estimate dopamine

increases. This enhances the robustness of model fitting as it only requires the amplitude of ΔSUVR and the time-to-peak of its derivative for fitting the ΔSUVR data.

The PET data appears to be exactly the same as in the previous Tomasi 2023 paper – ref 30. At a minimum this should be clearly stated. Figure 3D is the same data as Fig 7A in the previous paper. All 3 conditions of PET data and the data about ‘high’ have been previously published as part of the methods development paper. The fMRI data and analyses are new.

It is misleading to reference a paper to justify the PET methods used and then to use that same data in the current paper.

Response: We apologize for the misunderstanding on this point. In the cover letter we clearly stated that the PET data was previously validated in another manuscript and that the focus of the current manuscript is on the fMRI results, which are novel and have not been published. We also noted in the methods section that the PET data used for the dynamic analyses were based on the findings reported in Tomasi et al., 2023. We are surprised that it would be considered misleading to apply previously collected PET data for a completely new purpose: to investigate the relationship between dynamic dopamine changes and brain function assessed with fMRI. Some of the PET data and “high ratings” results were presented again here only because they were necessary for interpreting and contextualizing the novel fMRI results, which are the clear focus of this manuscript.

We have made these distinctions much clearer in the revised version (first paragraph of the methods, p. 10; as follows)

Some of the PET data and the behavioral data (‘high’ ratings) were previously reported in a recent publication (Tomasi et al., 2023). Here, we applied these data to investigate the relationship between dynamic dopamine changes and brain function assessed with fMRI; all of the primary fMRI results in this manuscript are novel and have not been published.

We have also explicitly noted which results were previously published (captions for Figures 2 and 3).

For inclusion/exclusion the authors did a commendable job excluding for other substance use and comorbidity, however nicotine/tobacco smoking was not mentioned even though use of nicotine has published effects on the DA system.

Response: All participants self-reported no history of nicotine/tobacco use; we have added this information to the Participants section in the methods.

Reviewer #3 (Remarks to the Author):

In this study, the researchers utilized a combined [11C]raclopride-PET/BOLD-fMRI technique to investigate the dynamics of dopamine release in the striatum and its correlation with overall brain activity and connectivity. They administered methylphenidate through both oral and intravenous routes

to achieve slow and rapid delivery to the brain. The findings showed that rapid dopamine increases triggered activation in the frontostriatal circuit, which includes the dorsal anterior cingulate cortex (dACC) and dorsal caudate. However, oral administration did not result in any activation in this circuit. Interestingly, the subjects reported high ratings corresponding to the observed effects which leads to the conclusion that dACC activation plays a key role in drug reward and may be a promising target for neuromodulations to treat substance use disorders.

Figure 2 shows [¹¹C]raclopride PET data on a temporal scale, which provides information about the rate and amplitude of DA (D2 binding decrease) in the striatum. This analysis approach does not allow for the examination of temporal fluctuations of DA in extrastriatal regions. Thanarajah et al.

(<https://pubmed.ncbi.nlm.nih.gov/30595479/>) recently related BOLD activations to temporal changes in extrasynaptic DA release in several brain regions using a [¹¹C]raclopride bolus infusion approach, and I believe this information would be very valuable for better interpreting the fMRI findings in the dACC.

Response: We agree that this would be a very valuable addition. However, Thanarajah et al. employed a PET-fMRI acquisition method sequentially, whereas we employed a simultaneous PET-MRI approach. The bolus+infusion technique was not feasible for us due to the challenges posed by the simultaneous PET-MRI setup. More specifically, to ensure safety, our magnetic Harvard pump had to be positioned 6 feet away from the MRI bore. Consequently, utilizing the bolus+injection method for ¹¹C-raclopride would have necessitated excessively high levels of initial radioactivity (>80mCi), which was deemed unsafe. Therefore we are unable to replicate their methodology. We now note this methodological limitation in the methods (footnote p. 14).

The present study demonstrates nonetheless a well-designed methodology and data analysis. The results section could benefit from additional analysis and a more comprehensive discussion of the findings.

Comments:

1. The hypothesis put forward by the authors in Figure 1c proposes an explanation for the fMRI signal changes based on the opposing effects of D2 and D1 receptors in the indirect and direct pathways. However, it is my belief that this hypothesis may not comprehensively elucidate the observed alterations fMRI and functional connectivity (FC) following pharmacological stimulations that enhances synaptic dopamine levels. Apart from the effects on postsynaptic receptors, there may also be presynaptic effects on D2 autoreceptors that inhibit phasic neuronal firing and dopamine release, leading to attenuated neuronal signaling. This attenuation could potentially account for the lack of response in the striatum.

Response: We agree that autoreceptor stimulation by dopamine is important to the observed findings. Our hypotheses was based on a prior model that had both simulations and empirical fMRI data in nonhuman primates to support it (Mandeville et al., 2013; Sander et al., 2013). Still, we now acknowledge that this is a simplified model based on postsynaptic dopamine receptor stimulation (figure caption 1) and have added the following discussion to p. 9:

Our hypothesis represented a simplified model that focused primarily on postsynaptic receptor stimulation. In general decreases in striatal D2R binding with [¹¹C]raclopride are interpreted to reflect predominantly postsynaptic D2R receptors since their relative concentration in striatum is estimated to be 4 times higher than that of D2 autoreceptors (Bello et al., 2011; Anzalone et al., 2012). Yet dopamine increases to MP are also likely to evoke presynaptic effects on D2 autoreceptors (Ford, 2014), which would have

attenuated peak dopamine increases by inhibiting dopamine cell firing and striatal dopamine release, ultimately attenuating the striatal BOLD responses.

2. The authors state in the discussion section 'striatal BOLD did not significantly correlate with dopamine dynamics in this study', since they observed no BOLD changes in the striatum. Did the authors also check the caudate nucleus and putamen? At least from supplemental figure S4, bottom (amplitude of dopamine), there seems to be decreased activity related to putamen in the placebo and iv condition, while the oral administration shows increased activity in the Caudate. In S7 (insular seeded FC), there seems to be decreased connectivity in the oral group between insular and putamen? Please provide any significant clusters of the BOLD signal changes in a table.

Response: We did not observe a significant association between rate of dopamine increases and BOLD in the caudate nor putamen; the results presented in Figure 2 reflect a whole-brain voxelwise analysis. The results from figure S4 and S7 are one-sample t-tests and are shown for completeness, but in the key analyses (paired t-tests between the MP conditions and placebo; Figure S5 and S9) we do not observe significant clusters within putamen. Significant clusters from all paired t-test results are now presented in Supplementary Tables 2-4.

3. Did the authors look at sex differences in their findings. Where the observed effects independent of the sex? I think this would be important to report even if there are no differences.

Response: Thanks for this suggestion; we have now performed these analyses. Results from two-sample t-tests showed no significant sex effects for behavior (max high ratings); nor time-to-peak dopamine increases estimated from PET data; nor baseline binding potential (BPnd) in the nucleus accumbens, caudate, or putamen; nor strength of association between rate of dopamine increases and fMRI activations from all significant clusters identified in the manuscript, even at an uncorrected $p < .05$ threshold (all p 's $> .22$). However, because this trial was not powered for sex effects analysis (11 males versus 9 females) we only report these null results as exploratory in the **Supplement**.

4. I would appreciate a more comprehensive explanation/discussion as to why the authors attribute the response to the intravenous drug administration to the dorsal (cognitive) region of the anterior cingulate cortex (ACC) rather than the rostral (affective) region. The ACC plays a significant role in various functions such as contextual memory, emotional regulation/appraisal, social cognition, resting-state default mode, and conflict monitoring. In the context of drug reward, what specific role does the ACC fulfill?

Response: The peak coordinate of this cluster (MNI: $x = -9$, $y = 12$, $z = 36$) certainly lies within the region commonly called 'dorsal' anterior cingulate cortex. Most of the significant cluster does not fall within the 'rostral' or 'subgenual' division, though there is no definitive parcellation of ACC and there is some debate as to where the boundaries of these subdivisions lie. Further, based on these resting state data we cannot definitively ascribe functional significance to the subregion of cingulate cortex that was significant in our analyses. Using the Neurosynth meta-analytic maps for "affective" and "cognitive control" terms, it appears that part of the 'cognitive control' map (blue) more readily falls near the peak of our significant cluster (red); in contrast most of the 'affective' map (green) lays rostrally outside of our cluster.

Still, this is only a descriptive look at these data and does not formally test for the functional significance of our cluster, and so we do not want to overinterpret these data. Nevertheless, we feel it is worth adding discussion on the role of the ACC in drug reward more generally. Here, we posit that the strong relationship between ACC signal and the subjective experience of drug effects reflects the ACC's role in interoceptive processing. We have added discussion on p. 8 as follows:

The ACC performs numerous roles in cognition, emotion, and reward (Bush et al., 2000, 2002), which could all be relevant for processing the acute effects of stimulant drugs (Breiter et al., 1997; Jenkins et al., 2004; Udo De Haes et al., 2007). However, subregions of ACC are differentially involved in reward-related processes. In contrast to rostral ACC, which has been implicated in (inhibitory) affective processing, dACC, which had the most overlap with our findings, has been implicated in higher-order (drug independent) reward-based decision making but also in the inhibition of drug craving (Zhao et al., 2012; Sullivan et al., 2022). However, given that our experiment was performed in resting state without any task, we posit that dACC activation may reflect interoceptive signals mediating the experience of drug reward (Paulus and Stewart, 2014). In fact, poor interoceptive awareness is a feature of substance use disorders and theorized to play a role in compulsive drug-seeking (Smith et al., 2021). Moreover, the subregion of the ACC that was significant in our analyses corresponds to the one that underlies the connectivity disruption associated with remission of addiction in patients suffering from brain lesions (Joutsa et al., 2022).

Minor comments:

1. Some groups have reported signal changes of [11C]raclopride-PET data in the cerebellum after pharmacological challenges. Could the authors provide the mean TAC files in the supplemental notes?

Response: We have now provided the mean TAC files, which demonstrate that the cerebellum was not responsive to the pharmacological interventions in this study. See also the figure below.

Figure: [11C]Raclopride Time Activity Curve (TAC) in the cerebellum (values normalized to a mean of 1 for each individual, to minimize individual differences in blood flow/metabolism and emphasize potential effects of Methylphenidate (MP) intervention). Oral MP was administered at time $T = 0$ min; The raclopride injection was administered at $T = 30$ min; and IV MP was administered at time $T = 60$ min.

2. In figure 2f, there is a BOLD signal increase of the placebo group after treatment. This is not discussed in the discussion or results.

Response: Though there appears to be a slight increase in vmPFC signal during the placebo session, this was not significant at threshold ($p < .001$ uncorrected, $p < .05$ cluster-correction), or even at a substantially lowered threshold of $p < .01$ uncorrected. Therefore we refrain from interpreting this pattern.

References

- Abreu ME, Bigelow GE, Fleisher L, Walsh SL (2001) Effect of intravenous injection speed on responses to cocaine and hydromorphone in humans. *Psychopharmacology* 154:76–84.
- Alpert NM, Badgaiyan RD, Livni E, Fischman AJ (2003) A novel method for noninvasive detection of neuromodulatory changes in specific neurotransmitter systems. *NeuroImage* 19:1049–1060.
- Anzalone A, Lizardi-Ortiz JE, Ramos M, Mei CD, Hopf FW, Iaccarino C, Halbout B, Jacobsen J, Kinoshita C, Welter M, Caron MG, Bonci A, Sulzer D, Borrelli E (2012) Dual Control of Dopamine Synthesis and Release by Presynaptic and Postsynaptic Dopamine D2 Receptors. *J Neurosci* 32:9023–9034.
- Bello EP, Mateo Y, Gelman DM, Noaín D, Shin JH, Low MJ, Alvarez VA, Lovinger DM, Rubinstein M (2011) Cocaine supersensitivity and enhanced motivation for reward in mice lacking dopamine D2 autoreceptors. *Nat Neurosci* 14:1033–1038.
- Bjarnadottir GD, Haraldsson HM, Rafnar BO, Sigurdsson E, Steingrimsson S, Johannsson M, Bragadottir H, Magnusson A (2015) Prevalent Intravenous Abuse of Methylphenidate Among Treatment-Seeking Patients With Substance Abuse Disorders: A Descriptive Population-Based Study. *Journal of Addiction Medicine* 9:188.
- Breiter HC, Gollub RL, Weisskoff RM, Kennedy DN, Makris N, Berke JD, Goodman JM, Kantor HL, Gastfriend DR, Riorden JP, Mathew RT, Rosen BR, Hyman SE (1997) Acute effects of cocaine on human brain activity and emotion. *Neuron* 19:591–611.
- Bush G, Luu P, Posner MI (2000) Cognitive and emotional influences in anterior cingulate cortex. *Trends in Cognitive Sciences* 4:215–222.
- Bush G, Vogt BA, Holmes J, Dale AM, Greve D, Jenike MA, Rosen BR (2002) Dorsal anterior cingulate cortex: A role in reward-based decision making. *Proceedings of the National Academy of Sciences* 99:523–528.
- Cortese S, D'Acunto G, Konofal E, Masi G, Vitiello B (2017) New Formulations of Methylphenidate for the Treatment of Attention-Deficit/Hyperactivity Disorder: Pharmacokinetics, Efficacy, and Tolerability. *CNS Drugs* 31:149–160.
- Crombag HS, Bossert JM, Koya E, Shaham Y (2008) Context-induced relapse to drug seeking: a review. *Philosophical Transactions of the Royal Society B: Biological Sciences* 363:3233–3243.
- Ford CP (2014) The role of D2-autoreceptors in regulating dopamine neuron activity and transmission. *Neuroscience* 282:13–22.
- Irace Z, Mérida I, Redouté J, Fonteneau C, Suaud-Chagny M-F, Brunelin J, Vidal B, Zimmer L, Reilhac A, Costes N (2020) Bayesian Estimation of the ntPET Model in Single-Scan Competition PET Studies. *Frontiers in Physiology* 11 Available at: <https://www.frontiersin.org/articles/10.3389/fphys.2020.00498> [Accessed July 10, 2023].
- Jenkins BG, Sanchez-Pernaute R, Brownell AL, Chen YCI, Isacson O (2004) Mapping dopamine function in primates using pharmacologic magnetic resonance imaging. *Journal of Neuroscience* 24:9553–9560.

- Joutsa J, Moussawi K, Siddiqi SH, Abdolahi A, Drew W, Cohen AL, Ross TJ, Deshpande HU, Wang HZ, Bruss J, Stein EA, Volkow ND, Grafman JH, van Wijnngaarden E, Boes AD, Fox MD (2022) Brain lesions disrupting addiction map to a common human brain circuit. *Nat Med* Available at: <https://www.nature.com/articles/s41591-022-01834-y> [Accessed June 15, 2022].
- Lambert NM, McLeod M, Schenk S (2006) Subjective responses to initial experience with cocaine: an exploration of the incentive-sensitization theory of drug abuse. *Addiction* 101:713–725.
- Lammertsma AA, Hume SP (1996) Simplified Reference Tissue Model for PET Receptor Studies. *NeuroImage* 4:153–158.
- Lau-Barraco C, Dunn ME (2009) Environmental context effects on alcohol cognitions and immediate alcohol consumption. *Addiction Research & Theory* 17:306–314.
- LeCocq MR, Randall PA, Besheer J, Chaudhri N (2020) Considering Drug-Associated Contexts in Substance Use Disorders and Treatment Development. *Neurotherapeutics* 17:43–54.
- Mandeville JB, Sander CYM, Jenkins BG, Hooker JM, Catana C, Vanduffel W, Alpert NM, Rosen BR, Normandin MD (2013) A receptor-based model for dopamine-induced fMRI signal. *NeuroImage* 75:46–57.
- Morton WA, Stock GG (2000) Methylphenidate Abuse and Psychiatric Side Effects. *Prim Care Companion J Clin Psychiatry* 02:159–164.
- Normandin MD, Schiffer WK, Morris ED (2012) A linear model for estimation of neurotransmitter response profiles from dynamic PET data. *NeuroImage* 59:2689–2699.
- Paulus MP, Stewart JL (2014) Interoception and drug addiction. *Neuropharmacology* 76:342–350.
- Resnick RB, Kestenbaum RS, Schwartz LK (1977) Acute Systemic Effects of Cocaine in Man: A Controlled Study by Intranasal and Intravenous Routes. *Science* 195:696–698.
- Sander CYM, Hooker JM, Catana C, Normandin MD, Alpert NM, Knudsen GM, Vanduffel W, Rosen BR, Mandeville JB (2013) Neurovascular coupling to D2/D3 dopamine receptor occupancy using simultaneous PET/functional MRI. *Proceedings of the National Academy of Sciences of the United States of America* 110:11169–11174.
- Smith R, Feinstein JS, Kuplicki R, Forthman KL, Stewart JL, Paulus MP, Khalsa SS (2021) Perceptual insensitivity to the modulation of interoceptive signals in depression, anxiety, and substance use disorders. *Sci Rep* 11:2108.
- Sullivan RM, Maple KE, Wallace AL, Thomas AM, Lisdahl KM (2022) Examining Inhibitory Affective Processing Within the Rostral Anterior Cingulate Cortex Among Abstinent Cannabis-Using Adolescents and Young Adults. *Frontiers in Psychiatry* 13 Available at: <https://www.frontiersin.org/articles/10.3389/fpsy.2022.851118> [Accessed June 29, 2023].
- Tomasi D, Manza P, Logan J, Shokri-Kojori E, Yonga M-V, Kroll D, Feldman D, McPherson K, Biessacker C, Dennis E, Johnson A, Yuan K, Wang W-T, Butman JA, Wang G-J, Volkow ND (2023) Time-varying SUVR reflects the dynamics of dopamine increases during methylphenidate challenges in humans. *Commun Biol* 6:1–10.

Udo De Haes JI, Maguire RP, Jager PL, Paans AMJ, Den Boer JA (2007) Methylphenidate-induced activation of the anterior cingulate but not the striatum: A [¹⁵O]H₂O PET study in healthy volunteers. *Human Brain Mapping* 28:625–635.

Volkow ND, Michaelides M, Baler R (2019) The Neuroscience of Drug Reward and Addiction. *Physiological Reviews* 99:2115–2140.

Zhao L-Y, Tian J, Wang W, Qin W, Shi J, Li Q, Yuan K, Dong M-H, Yang W-C, Wang Y-R, Sun L-L, Lu L (2012) The Role of Dorsal Anterior Cingulate Cortex in the Regulation of Craving by Reappraisal in Smokers. *PLOS ONE* 7:e43598.

REVIEWERS' COMMENTS

Reviewer #1 (Remarks to the Author):

The authors have addressed the comments I raised in the previous version. The one issue that remains concerns acknowledging the importance of drug-associated contexts and cues in modulating the dopamine response to drugs.

The text has been revised to state 'A rich literature in rats (Crombag et al., 2008) and humans (LeCocq et al., 2020) finds that the environmental context during drug exposure has a critical impact on subsequent drug use behavior'. Both of these references are reviews. In addition to these, please also include reference to original research demonstrating that drug-associated contexts and drug-associated cues are decisive in predicting the dopamine response to drugs.

In doing so, the authors would be remiss in not citing the work of the Leyton lab which has extensively characterized this effect in humans. As such, the following references should be cited :

Fotros, A. et al. (2013) Cocaine cue-induced dopamine re-lease in amygdala and hippocampus: a high-resolution PET [18F]fallypride study in cocaine dependent participants. *Neuropsychopharmacology* 38, 1780–1788

Cox, S.M.L. et al. (2017) Cocaine cue-induced dopamine release in recreational cocaine users. *Sci. Rep.* 7, 46665

Milella, M.S. et al. (2016) Cocaine cue-induced dopamine release in the human prefrontal cortex. *J. Psychiatry Neurosci.* 41, 322–330

Reviewer #2 (Remarks to the Author):

I appreciate the added information on why SUVr was chosen and the inclusion of data demonstrating a high correlation with other outcome measures.

One final point is that the response seems to imply that LSSRM/ntPET require a bolus + constant infusion paradigm, but that is not true. Those models can be applied to bolus data.

Thank you for making it clear to the reader what data has been previously published.

Reviewer #3 (Remarks to the Author):

The authors have fully addressed my comments. I would thus recommend the manuscript for publication.

We thank the reviewers again for their comments. Our responses are listed below.

Reviewer #1 (Remarks to the Author):

The authors have addressed the comments I raised in the previous version. The one issue that remains concerns acknowledging the importance of drug-associated contexts and cues in modulating the dopamine response to drugs.

The text has been revised to state 'A rich literature in rats (Crombag et al., 2008) and humans (LeCocq et al., 2020) finds that the environmental context during drug exposure has a critical impact on subsequent drug use behavior'. Both of these references are reviews. In addition to these, please also include reference to original research demonstrating that drug-associated contexts and drug-associated cues are decisive in predicting the dopamine response to drugs.

In doing so, the authors would be remiss in not citing the work of the Leyton lab which has extensively characterized this effect in humans. As such, the following references should be cited :

Fotros, A. et al. (2013) Cocaine cue-induced dopamine re-lease in amygdala and hippocampus: a high-resolution PET [18F]fallypride study in cocaine dependent participants. *Neuropsychopharmacology* 38, 1780–1788

Cox, S.M.L. et al. (2017) Cocaine cue-induced dopamine release in recreational cocaine users. *Sci. Rep.* 7, 46665

Milella, M.S. et al. (2016) Cocaine cue-induced dopamine release in the human prefrontal cortex. *J. Psychiatry Neurosci.* 41, 322–330

Response: Thank you for these suggestions. We now additionally cite several original research papers including the three suggested (refs 61-63 and 66-70).

Reviewer #2 (Remarks to the Author):

I appreciate the added information on why SUVr was chosen and the inclusion of data demonstrating a high correlation with other outcome measures.

One final point is that the response seems to imply that LSSRM/ntPET require a bolus + constant infusion paradigm, but that is not true. Those models can be applied to bolus data.

Response: This is true, to clarify we have now added the following to p. 14:

While LSSRM does not strictly require a paradigm with a continuous infusion, in our dataset we found that the relatively low radioactivity counts made dopamine quantification with LSSRM challenging.